# Online reinforcement learning of state representation in recurrent network supported by the power of random feedback and biological constraints

**Takayuki Tsurumi[1,2], Ayaka Kato[2,3], Arvind Kumar[2,4,5], Kenji Morita[1,2,6]***

[1]Physical and Health Education, Graduate School of Education, The University of Tokyo, Tokyo, Japan; [2]Theoretical Sciences Visiting Program, Okinawa Institute of Science and Technology, Okinawa, Japan; [3]Department of Psychiatry, Icahn School of Medicine at Mount Sinai, New York, United States; [4]Division of Computational Science and Technology, School of Electrical Engineering and Computer Science, KTH Royal Institute of Technology, Stockholm, Sweden; [5]Science for Life Laboratory, Solna, Stockholm, Sweden; [6]International Research Center for Neurointelligence (WPI-IRCN), The University of Tokyo, Tokyo, Japan

**\*For correspondence:**
morita@p.u-tokyo.ac.jp

## eLife Assessment

In this **important** study, the authors model reinforcement-learning experiments using a recurrent neural network. The work examines if the detailed credit assignment necessary for back-propagation through time can be replaced with random feedback. The authors provide **solid** evidence that the solution is adequate within relatively simple tasks.

**Abstract** Representation of external and internal states in the brain plays a critical role in enabling suitable behavior. Recent studies suggest that state representation and state value can be simultaneously learned through Temporal-Difference-Reinforcement-Learning (TDRL) and Backpropagation-Through-Time (BPTT) in recurrent neural networks (RNNs) and their readout. However, neural implementation of such learning remains unclear as BPTT requires offline update using transported downstream weights, which is suggested to be biologically implausible. We demonstrate that simple online training of RNNs using TD reward prediction error and random feedback, without additional memory or eligibility trace, can still learn the structure of tasks with cue–reward delay and timing variability. This is because TD learning itself is a solution for temporal credit assignment, and feedback alignment, a mechanism originally proposed for supervised learning, enables gradient approximation without weight transport. Furthermore, we show that biologically constraining downstream weights and random feedback to be non-negative not only preserves learning but may even enhance it because the non-negative constraint ensures loose alignment—allowing the downstream and feedback weights to roughly align from the beginning. These results provide insights into the neural mechanisms underlying the learning of state representation and value, highlighting the potential of random feedback and biological constraints.

## Introduction

Multiple lines of studies have suggested that Temporal-Difference-Reinforcement-Learning (TDRL) is implemented in the cortico-basal ganglia-dopamine (DA) circuits where DA encodes TD reward-prediction-error (RPE) (*Montague et al., 1996*; *Schultz et al., 1997*; *Niv and Schoenbaum, 2008*; *Cohen et al., 2012*; *Steinberg et al., 2013*; *Kim et al., 2020*) and DA-dependent plasticity of cortico-striatal synapses corresponds to TD-RPE-dependent update of state/action values (*Reynolds et al., 2001*; *Shen et al., 2008*; *Yagishita et al., 2014*). Traditionally, TDRL in the cortico-basal ganglia-DA circuits was considered to serve only for relatively simple behavior. However, subsequent studies suggested that more sophisticated, apparently goal-directed/model-based behavior can also be achieved by TDRL if states are appropriately represented (*Russek et al., 2017*; *Stachenfeld et al., 2017*; *Qian et al., 2025*) and that DA signals indeed reflect model-based predictions (*Langdon et al., 2018*; *Keiflin et al., 2019*). Conversely, impairments in state representation may relate to behavioral or mental health problems (*Redish et al., 2007*; *Gershman et al., 2013*; *Shimomura et al., 2021*; *Feng et al., 2021*; *Sato et al., 2023*). Early modeling studies treated state representations appropriate to the situation/task as given ('handcrafted' by the authors), but representation itself should be learned in the brain (*Gershman and Niv, 2010*; *Niv, 2019*; *George et al., 2023*; *Bono et al., 2023*; *Fang et al., 2023*; *Cone and Clopath, 2024*). Recently, it was shown that appropriate state representation can be learned through TDRL in a recurrent neural network (RNN) by minimizing squared TD value-error without explicit target (*Qian et al., 2025*; *Hennig et al., 2023*), while state value can be simultaneously learned in connections downstream of the RNN.

However, whether such a learning—referred to as the value-RNN (*Hennig et al., 2023*), can be implemented in the brain remains unclear. This is because the value-RNN (*Hennig et al., 2023*) used the Backpropagation-Through-Time (BPTT) (*Rumelhart et al., 1986b*), which applies gradient-descent error-'backpropagation' (hereafter referred to as backprop) (*Amari, 1967*; *Rumelhart et al., 1986a*) to a temporally unfolded RNNs. BPTT has been argued to be biologically implausible mainly due to problems with feedback and causality. Regarding feedback, updating upstream connections requires feedback whose weights are transported from downstream forward connections, but such weight transportation is difficult to implement biologically (*Grossberg, 1987*; *Crick, 1989*). In the case of the value-RNN, if the state-representing RNN and the value-encoding readout are implemented by the intra-cortical circuit and the striatal neurons, respectively, as generally assumed (*Montague et al., 1996*; *Doya, 2000*; *O'Doherty et al., 2004*), this weight transportation means that the update (plasticity) rule for intra-cortical connections involves the downstream cortico-striatal synaptic strengths, which are not accessible from the cortex. Regarding the problem of causality in BPTT, the error needs to be incrementally accumulated in the temporally backward order, but such an acausal offline update is biologically implausible (*Murray, 2019*; *Bellec et al., 2020*).

Recently, a potential solution for the feedback problem of backprop has been proposed (*Lillicrap et al., 2016*; see also *Guerguiev et al., 2017*; *Sacramento et al., 2018*; *Whittington and Bogacz, 2019*; *Lillicrap et al., 2020*; *Payeur et al., 2021*; *Greedy et al., 2022*; *Song et al., 2024*; *Pagkalos et al., 2024* for other approaches). Specifically, in supervised learning of feed-forward networks, it was shown that when the transported downstream weights used for updating upstream connections were replaced with fixed random strengths, comparable learning performance was still achieved (*Lillicrap et al., 2016*). This was suggested to be because the information of the random strengths transferred to the upstream connections and then to the downstream feed-forward connections so that these feed-forward connections became aligned to the random feedback and thereby the random feedback could work similarly to the feedback with transported downstream weights in backprop. This mechanism was named the 'feedback alignment' (FA) (*Lillicrap et al., 2016*), and was subsequently shown to work also in online supervised learning of RNN (algorithm named RFLO (random feedback local online)) (*Murray, 2019*) and proposed to be 'neurally' implemented (*Wärnberg and Kumar, 2023*) (in a different way from the present study as we discuss in the Discussion).

The value-RNN (*Qian et al., 2025*; *Hennig et al., 2023*) differs from supervised learning considered in these previous FA studies in two ways: (1) it is TD learning, that is, it approximates the true error by the TD-RPE because the true error, or true state value, is unknown, and (2) it uses a scalar error (TD-RPE) rather than a vector error. Scalar reward-based online learning of RNN with random feedback was actually shown to work in a different study (*Bellec et al., 2020*) (their Supplementary Figure 5), but TD-RPE was not introduced in that setup, while the same study also examined another

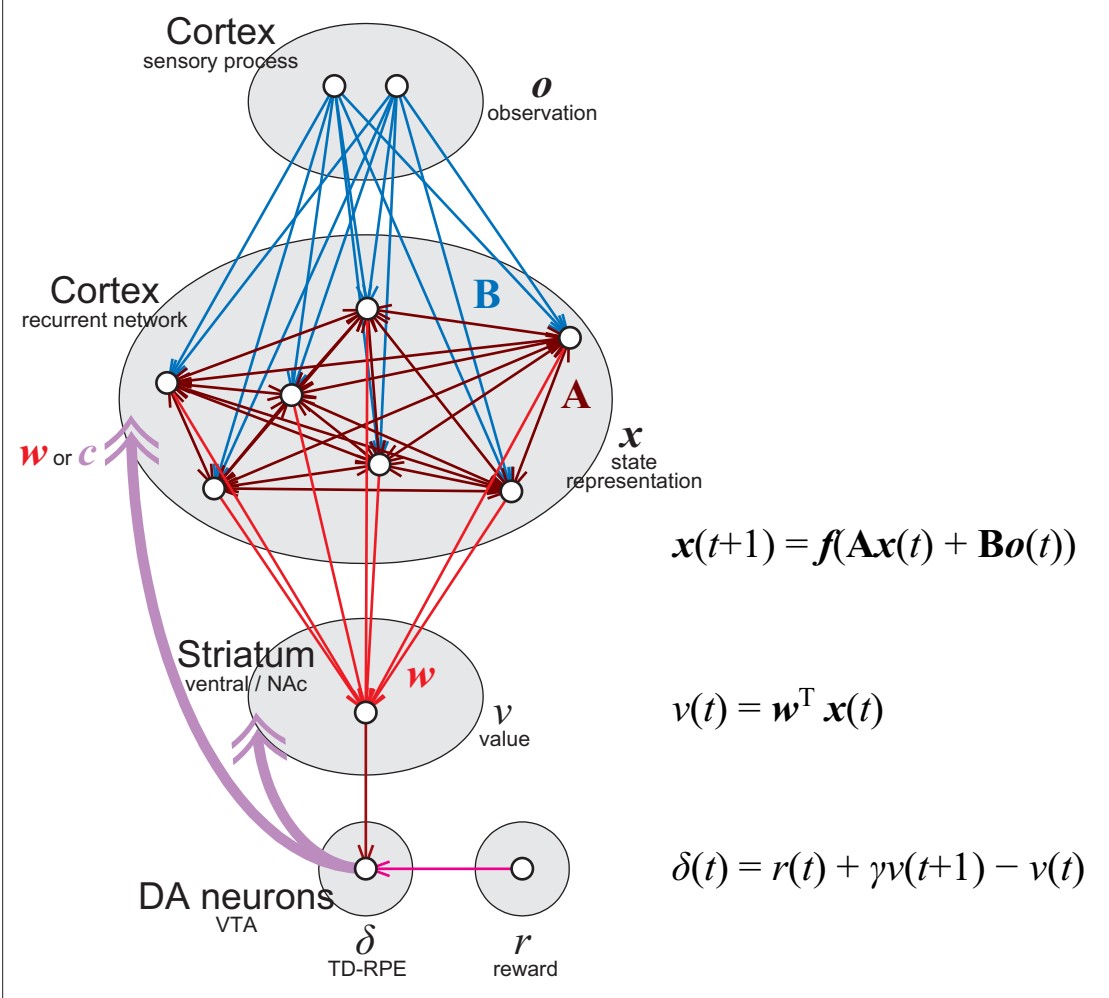

**Figure 1.** Implementation of the online value-recurrent neural network (RNN) in the cortico-basal ganglia-DA circuits.

setup with TD-RPE, but the result with random feedback was not shown for this latter setup (their Figures 4 and 5). Therefore, it was non-trivial whether the value-RNN could be modified to incorporate online update using random feedback. Here, we demonstrate that such a modified value-RNN could still work and provide a mechanistic insight into how it works.

Next, we address other biological plausibility issues with FA-based rule. Specifically, we imposed biological constraints that the downstream (cortico-striatal) weights and the fixed random feedback, as well as the activities of neurons in the RNN, were all non-negative. Moreover, we also remedied the non-monotonic dependence of the update of RNN connection strength on post-synaptic neural activity. We found that the non-negative constraint appeared to aid, rather than degrade, the learning by ensuring that the downstream weights and the fixed random feedback were loosely aligned from the beginning. These results suggest how learning of state representation and value could be implemented via DA-dependent synaptic plasticity in cortical and striatal circuits, where DA encodes TD-RPE.

## Results
### Online value-RNN with fixed random feedback

We considered an online value-RNN in the cortico-basal ganglia circuits (*Figure 1*). In this model, a cortical region/population represents information of sensory observation (*o*) and sends it to another cortical region/population which estimates a state (*x*) given the sensory inputs. We approximate this cortical population as an RNN (number of RNN units was varied between 5 and 40). Neurons in the

RNN learn to represent states by updating the strengths of recurrent connections **A** and feed-forward connections **B**. The activity of a population of striatal neurons that receive inputs from the RNN is supposed to learn to represent the state values (*v*), by learning the weights (*w*) of cortico-striatal connections. DA neurons in the ventral tegmental area (VTA) receive information about the value and reward (*r*) from the striatum (both direct and indirect pathways) and other structures, and the activity of the DA neurons, as well as released DA, represents TD-RPE ($\delta$). The TD-RPE-representing DA is released in the striatum and also in the cortical RNN through mesocorticolimbic projections and used for modifying **A**, **B**, and *w* (*Figure 1*).

The original value-RNN (*Qian et al., 2025*; *Hennig et al., 2023*) adopted BPTT (*Rumelhart et al., 1986b*) as an update rule for the connections onto the RNN (**A** and **B**), which requires the (gradually changing) value weights (*w*), but this is biologically implausible because the cortico-striatal synaptic weights are not available in the cortex as mentioned above. Therefore, we examined a model (agent) in which these weights were replaced with fixed random strengths (*c*), in comparison with a model that used these weights. Because the temporally acausal offline update used in BPTT is also biologically implausible, we used an online learning rule, which considers only the influence of the recurrent weights at the previous time step (see the Methods for details and equations). We refer to these models (agents) as the online value-RNN or oVRNN.

We assumed that a single RNN unit corresponds to a small population of neurons that intrinsically share inputs and outputs, for genetic or developmental reasons, and the activity of each unit represents the (relative) firing rate of the population. Cortical population activity is suggested to be sustained not only by fast synaptic transmission and spiking but also, even predominantly, by slower synaptic neurochemical dynamics (*Mongillo et al., 2008*) such as short-term facilitation, whose time constant can be around 500 ms (*Morishima et al., 2011*). Therefore, we assumed that a single time step of our rate-based (rather than spike-based) model corresponds to 500 ms.

In each simulation, the recurrent (**A**) and feed-forward connection (**B**) weights onto the RNN units were initialized to pseudo standard normal random numbers. As a negative control, we also conducted simulations in which these connections were not updated from initial values, referring to as the case with 'untrained (fixed) RNN'. Notably, the value weights *w* (i.e., connection weights from the RNN to the striatal value unit) were still trained in the models with untrained RNN. The oVRNN models, and the model with untrained RNN, were continuously trained across trials in each task, because we considered that it was ecologically more plausible than episodic training of separate trials.

## Simulation of a Pavlovian cue–reward association task with variable inter-trial intervals

First, we took a small RNN with 7 units to represent state in the cortex and simulated a Pavlovian cue–reward association task, in which a cue was followed by a reward three time steps later, and inter-trial interval (ITI, i.e., reward to next cue) was randomly chosen from 4, 5, 6, or 7 time steps (*Figure 2A*). Given that a single time step corresponds to 500 ms as mentioned above, three time steps from cue to reward correspond to 1.5 s, which matches the delay in the conditioning task used by *Schultz et al., 1997*. In this task, states after receiving cue information can be defined by time steps from the cue, and the state values of these states can be estimated by calculating the expected cumulative discounted future rewards (*Sutton and Barto, 2018*) through simulations; we refer to them as 'estimated true state values' (*Figure 2B*, black line). Expected TD-RPE can be calculated from these estimated true values (*Figure 2B*, red line).

First, for comparison, we examined the traditional TD-RL agent with punctate state representation (without using the RNN), in which each state (time step from a cue) was represented in a punctate manner, that is, by a one-hot vector such as (1, 0,..., 0), (0, 1,..., 0), and so on. We examined two cases: one in which training was done in an episodic manner without continuation between trials (i.e., the value of the last state in a trial was not updated by TD-RPE upon entering the next trial) and the other in which training was done continuously across trials, as in the cases of agents using the RNN. The former agent developed positive values between cue and reward, and abrupt TD-RPE upon cue (*Figure 2C*), whereas the latter agent developed positive values also for states in the ITI (*Figure 2D*), looking similar to the true values (*Figure 2B*).

We then examined our oVRNN agents, with backprop-type transported downstream weights (oVRNNbp: *Figure 2E*) or with fixed random feedback (oVRNNrf: *Figure 2F*), in comparison with the

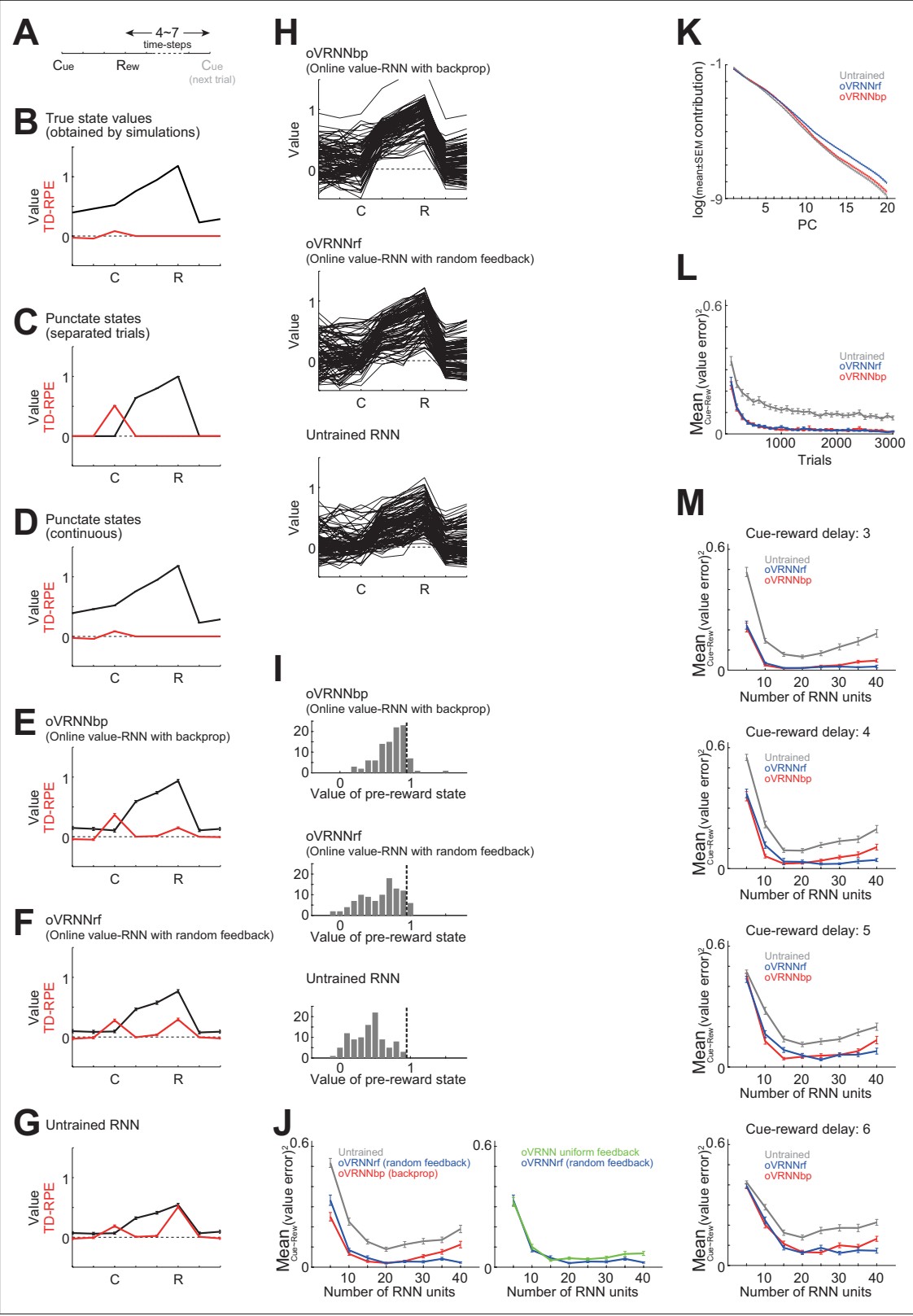

**Figure 2.** Simulation of a Pavlovian cue–reward association task. (**A**) Simulated task with variable inter-trial intervals (ITIs). (**B**) Black line: Estimated true values of states/timings through simulations according to the definition of state value, that is, expected cumulative discounted future rewards, taking into account the effect of probabilistic ITI. Red line: TD-RPEs calculated from the estimated true state/timing values. (**C–G**) State values (black lines) and TD-RPEs (red lines) at 1000th trial, averaged across 100 simulations (error bars indicating mean ± SEM across simulations; same applied to the followings

*Figure 2 continued on next page*

*Figure 2 continued*

unless otherwise mentioned), in different types of agent: (**C**) TD-RL agent having punctate state representation and state values without continuation between trials (i.e., the value of the last state in a trial was not updated by TD-RPE upon entering the next trial); (**D**) TD-RL agent having punctate state representation and continuously updated state values across trials; (**E**) Online value-recurrent neural network (RNN) with backprop (oVRNNbp). The number of RNN units was 7 (same applied to **F, G**); (**F**) Online value-RNN with fixed random feedback (oVRNNrf); (**G**) Agent with untrained RNN. (**H**) State values at 1000th trial in individual simulations of oVRNNbp (top), oVRNNrf (middle), and untrained RNN (bottom). (**I**) Histograms of the value of the pre-reward state (i.e., the state one time step before the reward state) at 1000th trial in individual simulations of the three models. The vertical black dashed lines indicate the true value of the pre-reward state (estimated through simulations). (**J**) *Left*: Mean of the squares of differences between the state values developed by each agent and the estimated true state values between cue and reward (referred to as the mean squared value-error) at 1000th trial in oVRNNbp (red line), oVRNNrf (blue line), and the model with untrained RNN (gray line) when the number of RNN units ($n$) was varied from 5 to 40. Learning rate for value weights was normalized by dividing by $n/7$ (same applied to the followings unless otherwise mentioned). *Right*: Mean squared value-error in oVRNNrf (blue line: same data as in the left panel) and oVRNN with uniform feedback (green line). (**K**) Log contribution ratios of the principal components of the time series (for 1000 trials) of RNN activities in each model with 20 RNN units. (**L**) Mean squared value-error in each model with 20 RNN units across trials. (**M**) Mean squared value-error in each model at 3000th trial in the cases where the cue–reward delay was 3, 4, 5, or 6 time steps (top to bottom panels).

agent with untrained fixed RNN (***Figure 2G***). As shown in the figures, oVRNNbp successfully learned the values of states between cue and reward, and oVRNNrf also learned these values, although to a somewhat smaller degree on average. On the other hand, the agent with untrained RNN developed the smallest state values on average among the three agents. This inferiority of untrained RNN may sound odd because there were only four states from cue to reward while random RNN with enough units is expected to be able to represent many different states (cf., ***Rajan and Abbott, 2006***) and the effectiveness of training of only the readout weights has been shown in reservoir computing studies (***Dominey, 1995***; ***Maass et al., 2002***; ***Jaeger, 2007***; ***Tanaka et al., 2019***). However, there was a difficulty stemming from the continuous training across trials (rather than episodic training of separate trials): the activity of untrained RNN upon cue presentation generally differed from trial to trial, and so it is non-trivial that cue presentation in different trials should be regarded as the same single state, even if it could eventually be dealt with at the readout level if the number of units increases.

The results above indicate that value-RNN could be trained online by fixed random feedback at least to a certain extent, although somewhat less effectively than by backprop-type feedback. Results of individual simulations shown in ***Figure 2H, I*** indicate that state values developed in oVRNNrf were largely comparable to those developed in oVRNNbp once they were successfully learned, but the success rate was smaller than oVRNNbp while still larger than the untrained RNN.

## Systematic simulations and analyses

Next, we tested whether learning performance of oVRNNbp, oVRNNrf, and the agent with untrained fixed RNN depends on the number of RNN units ($n$). For a valid comparison with the previously shown cases with 7 RNN units, the learning rate for the value weights was normalized by dividing by $n/7$. Learning performance was measured by the mean of squares of differences between the state values developed by each of these three types of agents and the estimated true state values (***Figure 2B***) between cue and reward at 1000th trial. As shown in the left panel of ***Figure 2J***, on average across simulations, oVRNNbp and oVRNNrf exhibited largely comparable performance and always outperformed the untrained RNN ($p < 0.00022$ in Wilcoxon rank sum test for oVRNNbp or oVRNNrf vs untrained for each number of RNN units), although oVRNNbp somewhat outperformed or underperformed oVRNNrf when the number of RNN units was small ($\leq 10$ ($p < 0.049$)) or large ($\geq 25$ ($p < 0.045$)), respectively. As the number of RNN units increased from 5 to 15 or 20, all three agents improved their performance. Additional increase of RNN units did not largely change the mean performance in oVRNNrf, while moderately decreasing it in oVRNNbp and untrained RNN. The green line in ***Figure 2J***, right shows the performance of a special case where the random feedback in oVRNNrf was fixed to the direction of $(1, 1,..., 1)^T$ (i.e., uniform feedback) with a random coefficient, which was largely comparable to, but somewhat worse than, that for the general oVRNNrf (blue line).

In order to examine the dimensionality of RNN dynamics, we conducted principal component analysis (PCA) of the time series (for 1000 trials) of RNN activities and calculated the contribution ratios of PCs in the cases of oVRNNbp, oVRNNrf, and untrained RNN with 20 RNN units. ***Figure 2K*** shows a log of contribution ratios of 20 PCs in each case. Compared with the case of untrained RNN, in oVRNNbp and oVRNNrf, initial component(s) had smaller contributions (PC1 ($t$-test $p = 0.00018$

in oVRNNbp; p = 0.0058 in oVRNNrf) and PC2 (p = 0.080 in oVRNNbp; p = 0.0026 in oVRNNrf)) while later components had larger contributions (PC3–10, 15–20, p < 0.041 in oVRNNbp; PC5–20, p < 0.0017 in oVRNNrf) on average, and this is considered to underlie their superior learning performance. We noticed that late components had larger contributions in oVRNNrf than in oVRNNbp, although these two models with 20 RNN units were comparable in terms of cue–reward state values (*Figure 2J*, left).

We examined how learning proceeded across trials in the models with 20 RNN units. As shown in *Figure 2L*, learning became largely converged by the 1000th trial, although slight improvement continued afterward. We further examined the cases with longer cue–reward delays. As shown in *Figure 2M*, as the delay increased, the mean squared error of state values (at 3000-th trial) increased, but the relative superiority of oVRNNbp and oVRNNrf over the model with untrained RNN remained to hold, except for cases with small number of RNN units (5) and long delay (5 or 6) (p < 0.0025 in Wilcoxon rank sum test for oVRNNbp or oVRNNrf vs untrained for each number of RNN units for each delay).

## Occurrence of FA and an intuitive understanding of its mechanism

Next, we questioned how FA contributes to the learnability of oVRNNrf. To address this question, we used an RNN with 7 units and examined whether the value weight vector $w$ became aligned to the random feedback vector $c$ in oVRNNrf, by looking at the changes in the angle between these two vectors across trials. As shown in *Figure 3A*, this angle, averaged across simulations, decreased over trials, indicating that the value weight $w$ indeed tended to become aligned to the random feedback $c$. We then examined whether better alignment of $w$ to $c$ related to better development of state value by looking at the relation between the angle between $w$ and $c$ and the value of the pre-reward state at 1000th trial. As shown in *Figure 3B*, there was a negative correlation such that the smaller the angle (i.e., more aligned), the larger the state value ($r = -0.288$, p = 0.00362), consistent with our expectation. These results indicate that the mechanism of FA, previously shown to work for supervised learning, also worked for TD learning of value weights and recurrent/feed-forward connections.

How did the FA mechanistically occur? We made an attempt to obtain an intuitive understanding. Assume that positive TD-RPE ($\delta(t) > 0$) is generated in a state, $S(= x(t))$ in a trial. Because of the update rule for $w$($w \leftarrow w + a\delta(t)x(t)$) (*Equation 1.9* in the Methods), $w$ is updated in the direction of $x(t)$. Next, what is the effect of updates of recurrent/feed-forward connections (**A** and **B**) on $x$? For simplicity, here we consider the case where observation is null ($o = \mathbf{0}$) and so $x(t) = f(Ax(t-1))$ holds (but a similar argument can be done in the case where observation is not null). If **A** is replaced with its updated one, it can be calculated that the $i$th element of $Ax(t-1)$ will hypothetically change by $c_i \times$ (a positive value) (technical note: the value is $a\delta(t)\{\Sigma_j x_j(t-1)^2\}(0.5 + x_i(t))(0.5 - x_i(t))$ (cf., *Equation 1.10*) which is positive unless $x(t-1) = \mathbf{0}$), and therefore the vector $Ax(t-1)$ as a whole will hypothetically change by a vector that is in a relatively close angle with $c$, or more specifically, is in the same quadrant as (and thus within at maximum 90° from) $c$ (e.g, $[c_1\ c_2\ c_3]^T$ and $[0.5c_1\ 1.2c_2\ 0.8c_3]^T$). Then, because $f$ is a monotonically increasing sigmoidal function, $x(t) = f(Ax(t-1))$ will also hypothetically change by a vector that is in a relatively close angle with $c$. This was indeed the case in our simulations as shown in *Figure 3C*, which plotted the angle between the hypothetical change in $x(t) = f(Ax(t-1), Bo(t-1))$ in cases **A** and **B** were replaced with their updated ones, multiplied with the sign of TD-RPE (sign($\delta(t)$)), and the fixed random feedback vector $c$ across time steps.

In this way, at state $S$ where TD-RPE is positive, $w$ is updated in the direction of $x(t)$, and $x(t)$ will hypothetically change by a vector that is in a relatively close angle with $c$ if **A** is replaced with its updated one. Then, if the update of $w$ and the hypothetical change in $x(t)$ due to the update of **A** could be integrated, $w$ would become aligned to $c$ (if TD-RPE is instead negative, $w$ is updated in the opposite direction of $x(t)$, and $x(t)$ will hypothetically change by a vector that is in a relatively close angle with $-c$, and so the same story holds in the end).

There is, however, a caveat regarding how the update of $w$ and the hypothetical change in $x(t)$ can be integrated. Although technical, here we briefly describe the caveat and a possible solution for it. The updates of $w$ and **A** use TD-RPE, which are calculated based on $v(t) = w^T x(t)$ and $v(t+1) = w^T x(t+1)$, and so $x(t)$ and $x(t+1)$ should already be determined beforehand. Therefore, the hypothetical change in $x(t)$ due to the update of **A**, described in the above, does not actually occur (this was why we mentioned 'hypothetical') and thus cannot be integrated with the update of $w$. Nevertheless,

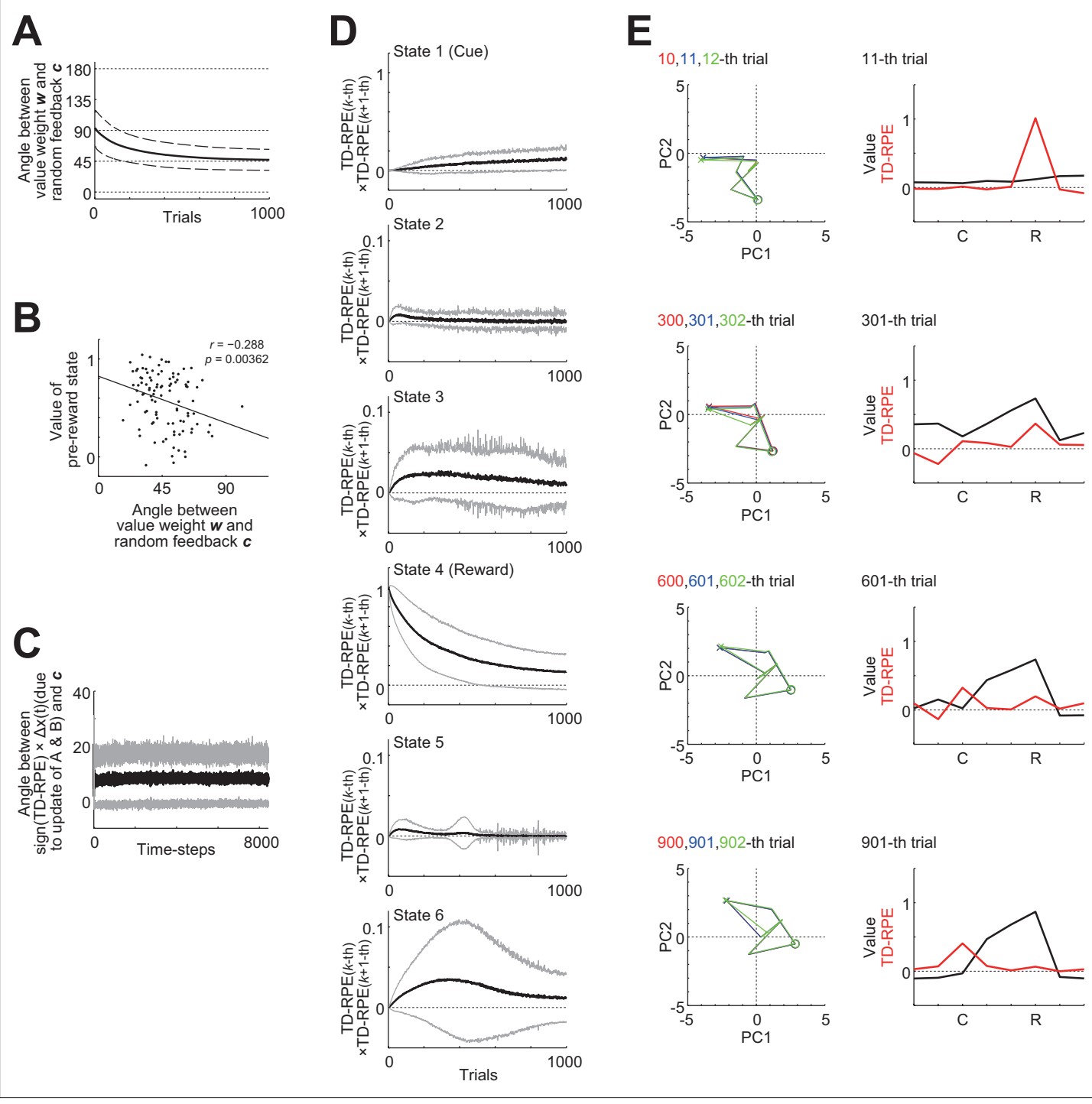

**Figure 3.** Occurrence of feedback alignment and an intuitive understanding of its mechanism. (**A**) Over-trial changes in the angle between the value-weight vector *w* and the fixed random feedback vector *c* in the simulations of oVRNNrf (7 recurrent neural network [RNN] units). The solid line and the dashed lines indicate the mean ± SD across 100 simulations, respectively. (**B**) Negative correlation (*r* = −0.288, *p* = 0.00362) between the angle between *w* and *c* (horizontal axis) and the value of the pre-reward state (vertical axis) at 1000th trial. The dots indicate the results of individual simulations, and the line indicates the regression line. (**C**) Angle between the hypothetical change in $x(t) = f(Ax(t-1), Bo(t-1))$ in cases **A** and **B** were replaced with their updated ones, multiplied with the sign of TD-RPE (sign(δ(*t*))), and the fixed random feedback vector *c* across time steps. The black thick line and the gray lines indicate the mean ± SD across 100 simulations, respectively (same applied to (**D**)). (**D**) Multiplication of TD-RPEs in successive trials at individual states (top: cue, fourth from the top: reward). Positive or negative value indicates that TD-RPEs in successive trials have the same or different signs, respectively. (**E**) *Left*: RNN trajectories mapped onto the primary and secondary principal components (horizontal and vertical axes, respectively) in three successive trials (red, blue, and green lines (heavily overlapped)) at different phases in an example simulation (10th to 12th, 300th to 302nd, 600th

*Figure 3 continued on next page*

*Figure 3 continued*

to 602nd, and 900th to 902nd trials from top to bottom). The crosses and circles indicate the cue and reward states, respectively. *Right*: State values (black lines) and TD-RPEs (red lines) at 11th, 301st, 601st, and 901st trials.

integration could still occur across successive trials, at least to a certain extent. Specifically, although TD-RPEs at $S$ in successive trials would generally differ from each other, they would still tend to have the same sign, as was indeed the case in our simulations (*Figure 3D*). Also, although the trajectories of RNN activity ($x$) in successive trials would differ, we could expect a certain level of similarity because the RNN is entrained by observation-representing inputs, again as was indeed the case in our example simulation (*Figure 3E*). Then, the hypothetical change in $x(t)$ due to the update of $\mathbf{A}$, considered above, could become a reality in the next trial, to a certain extent, and could thus be integrated into the update of $w$, explaining the occurrence of FA.

## Simulation of tasks with probabilistic structures of reward timing/existence

Previous work (*Starkweather et al., 2017*) examined the response of DA neurons in cue–reward association tasks in which reward timing was probabilistically determined (early in some trials but late in other trials). There were two tasks, which were largely similar, but there was a key difference that reward was given in all the trials in one task, whereas reward was omitted in some randomly determined trials in another task. *Starkweather et al., 2017* found that the DA response to later reward was smaller than the response to earlier reward in the former task, presumably reflecting the animal's belief that delayed reward will surely come, but the opposite was the case in the latter task, presumably because the animal suspected that reward was omitted in that trial. *Starkweather et al., 2017* then showed that such response patterns could be explained if DA encoded TD-RPE under particular state representations that incorporated the probabilistic structures of the task (called the 'belief state'). In that study, such state representations were 'handcrafted' by the authors, but the subsequent work (*Hennig et al., 2023*) showed that the original value-RNN with backprop (BPTT) could develop similar representations and reproduce the experimentally observed DA patterns.

In order to examine if our online value-RNN with fixed random feedback could also explain those experimental results, we simulated two tasks (*Figure 4A*) that were qualitatively similar to (though simpler than) the two tasks examined in the experiments (*Starkweather et al., 2017*). In our task 1, a cue was always followed by a reward either two or four time steps later with equal probabilities. Task 2 was the same as task 1 except that reward was omitted with 40% probability. In task 1, if reward was not given at the early timing (i.e., two steps later than cue), agent could predict that reward would be given at the late timing (i.e., four steps later than cue), and thus TD-RPE upon reward at the late timing is expected to be smaller than TD-RPE upon reward at the early timing (if agent perfectly learned the task structure, TD-RPE upon reward at the late timing should be 0). By contrast, in task 2, if reward was not given at the early timing, it might indicate that reward would be given at the late timing but might instead indicate that reward would be omitted in that trial, and thus TD-RPE upon reward at the late timing is expected to exist and can even be larger than TD-RPE upon reward at the early timing.

In these tasks, states can be defined in the following way. There were two types of trials, with early or late reward, in task 1, and additionally one more type of trial, without reward, in task 2 (*Figure 4Ba*, top). For each timing after receival of cue information in each of these trial types, its value can be estimated through simulations (*Figure 4Ba*, bottom). The agent could not know the current trial type until receiving reward at the early timing or the late timing or receiving no reward at both timings. Until these timings, the agent could have probabilistic belief about the current trial type, for example, 50% in the trial with early reward and 50% in the trial with late reward (in task 1) or 30% in the trial with early reward, 30% in the trial with late reward, and 40% in the trial without reward (in task 2) (*Figure 4Bb*). States of the timings after receival of cue information can be defined by incorporating these probabilistic beliefs at each timing (*Figure 4Bc*, top). Their true values can be calculated by taking (mathematical) expected value of the estimated values of each timing in each trial type (*Figure 4Bc*, bottom). Expected TD-RPE calculated from the estimated true values (*Figure 4C*) exhibited features that matched the conjecture mentioned above: in task 1, TD-RPE upon reception of late reward, which was actually 0, was smaller than TD-RPE upon reception of early reward, whereas in task 2, TD-RPE upon reception of late reward was larger than TD-RPE upon reception of early reward (as indicated by the inequality signs on *Figure 4C*).

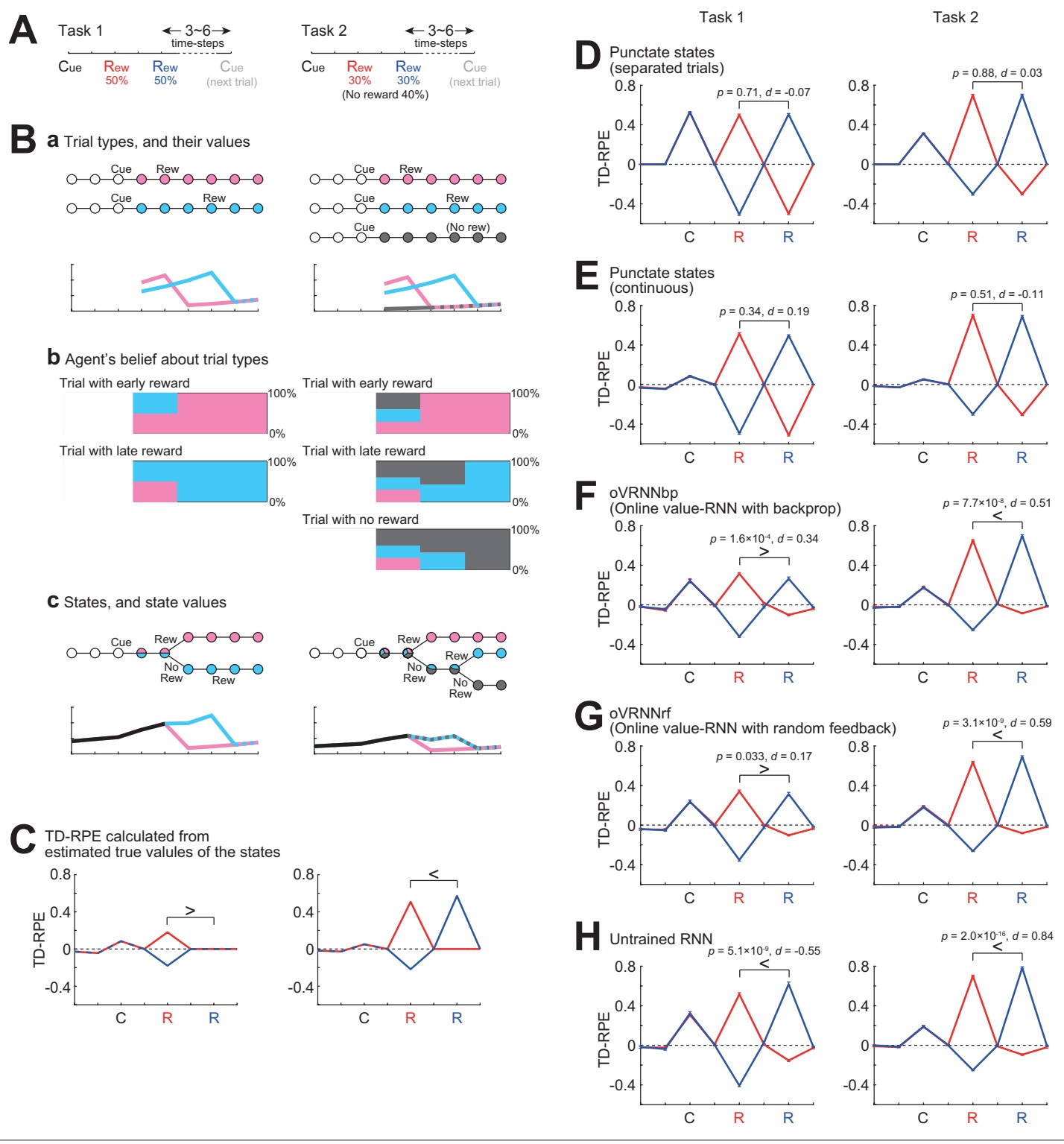

**Figure 4.** Simulation of two tasks having probabilistic structures, which were qualitatively similar to the two tasks examined in experiments (*Starkweather et al., 2017*) and modeled by the original value-recurrent neural network (RNN) with Backpropagation-Through-Time (BPTT) (*Hennig et al., 2023*). (**A**) Simulated two tasks, in which reward was given at the early or the late timing with equal probabilities in all the trials (task 1) or 60% of trials (task 2). (**B**) (**a**) *Top*: Trial types. Two trial types (with early reward and with late reward) in task 1 and three trial types (with early reward, with late reward, and without reward) in task 2. *Bottom*: Value of each timing in each trial type estimated through simulations. (**b**) Agent's probabilistic belief about the current trial type, in the case where agent was in fact in the trial with early reward (top row), the trial with late reward (second row), or the trial

*Figure 4 continued on next page*

*Figure 4 continued*

without reward (third row in task 2). (**c**) *Top*: States defined by considering the probabilistic beliefs at each timing from cue. *Bottom*: True state/timing values calculated by taking (mathematical) expected value of the estimated value of each timing in each trial type. (**C**) Expected TD-RPE calculated from the estimated true values of the states/timings for task 1 (left) and task 2 (right). Red lines: case where reward was given at the early timing, blue lines: case where reward was given at the late timing. It is expected that TD-RPE at early reward is larger than TD-RPE at late reward in task 1, whereas the opposite is the case in task 2, as indicated by the inequality signs. (**D–H**) TD-RPEs at the latest trial within 1000 trials in which reward was given at the early timing (red lines) or the late timing (blue lines), averaged across 100 simulations (error bars indicating ± SEM across simulations), in the different types of agent: TD-RL agent having punctate state representation and state values without (**D**) or with (**E**) continuation between trials; (**F**) oVRNNbp. The number of RNN units was 12 (same applied to (**G, H**)); (**G**) oVRNNrf; (**H**) agent with untrained RNN. The p values are for paired *t*-test between TD-RPE at early reward and TD-RPE at late reward (100 pairs, two-tailed), and the *d* values are Cohen's *d* using an average variance and their signs are with respect to the expected patterns shown in (**C**) (same applied to Figures 8 and 9C).

As mentioned above, the previous work (*Starkweather et al., 2017*) has shown that VTA DA neurons exhibited similar activity patterns to the abovementioned TD-RPE patterns, and the subsequent work (*Hennig et al., 2023*) has shown that the original value-RNN with backprop (BPTT) could reproduce such TD-RPE patterns. We examined how our oVRNNbp and oVRNNrf (with 12 RNN units) behaved in our simulated two tasks. oVRNNbp developed the expected TD-RPE patterns, that is, smaller TD-RPE upon late than early timing in task 1 but opposite pattern in task 2 (*Figure 4F*), and oVRNNrf also developed such patterns, although the effect size for task 1 was small (*Figure 4G*). These results indicate that online value-RNN could learn the probabilistic structures of the tasks even with fixed random feedback. By contrast, agents with punctate state representation without or with continuous value update across trials (*Figure 4D, E*), as well as agents with untrained fixed RNN (*Figure 4H*), could not develop such patterns well.

## Online value-RNN with further biological constraints

So far, the activities of neurons in the RNN ($x$) were initialized to pseudo standard normal random numbers, and thereafter took numbers in the range between −0.5 and 0.5 that was the range of the sigmoidal input-output function. The value weights ($w$) could also take both positive and negative values since no constraint was imposed. The fixed random feedback in oVRNNrf ($c$) was generated by pseudo standard normal random numbers, and so could also be positive or negative. Negativity of the neurons' activities and the value weights could potentially be regarded as inhibitory or smaller-than-baseline quantities. However, because neuronal firing rate is non-negative and cortico-striatal projections are excitatory, it would be biologically more plausible to assume that the activities of neurons in the RNN and the value weights are non-negative. As for the fixed random feedback, if it is negative, the update rule becomes anti-Hebbian under positive TD-RPE, and so assuming non-negativity would be plausible since Hebbian property has been suggested for rapid plasticity of cortical synapses (*Feldman, 2009*) (see Appendix 1.2 for possible consideration of behavioral time-scale synaptic plasticity (BTSP) in our models). Regarding the connection weights in/onto the RNN, here we keep the original assumption that they could be positive or negative because it could be an approximate description of recurrent neuronal network with both recurrent excitation and inhibition. Later (the 'Models with excitatory and inhibitory units' section) we will examine extended models that incorporate excitatory and inhibitory units and conform to Dale's law.

Other than the sign of connection weights, there was another biological plausibility issue in the update rule for recurrent and feed-forward connections that were derived from the gradient descent. Specifically, the dependence on the post-synaptic activity was non-monotonic, maximized at the middle of the range of activity. It would be more biologically plausible to assume a monotonic increase (while an *opposite* shape of non-monotonicity, once decrease and thereafter increase, called the BCM (Bienenstock–Cooper–Munro) rule has actually been suggested; *Bienenstock et al., 1982*; *Gjorgjieva et al., 2011*; *Shouval, 2011*).

In order to address these issues, we considered revised models. We first considered a revised oVRNNbp (with backprop-type transported weights), referred to as oVRNNbp-rev, in which the RNN activities and the value weights were constrained to be non-negative, while the non-monotonic dependence of the update rule on the post-synaptic activity remained unchanged (*Figure 5A*). We then considered a revised oVRNNrf, referred to as oVRNNrf-bio, in which the fixed random feedback, as well as the RNN activities and the value weights, were constrained to be non-negative, and also the

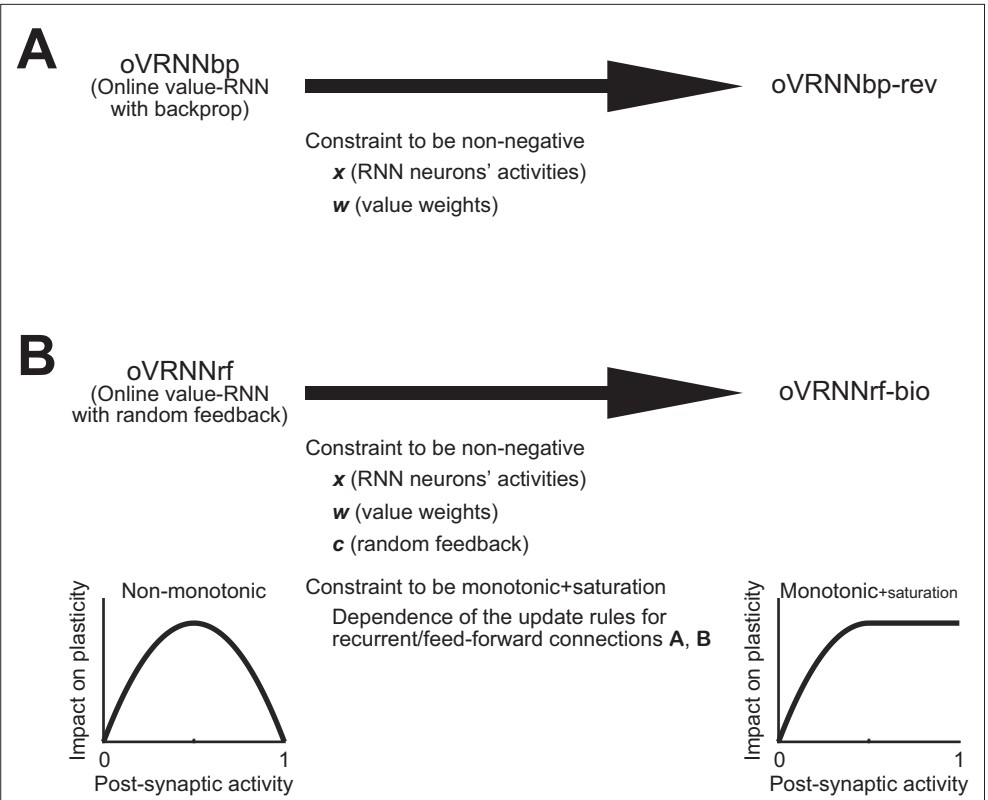

**Figure 5.** Revised online value-recurrent neural network (RNN) models with further biological constraints. (**A**) oVRNNbp-rev: oVRNNbp (online value-RNN with backprop) was modified so that the activities of neurons in the RNN (*x*) and the value weights (*w*) became non-negative. (**B**) oVRNNrf-bio: oVRNNrf (online value-RNN with fixed random feedback) was modified so that *x* and *w*, as well as the fixed random feedback (*c*), became non-negative and also the dependence of the update rules for recurrent/feed-forward connections (**A, B**) on post-synaptic activity became monotonic + saturation.

update rule was modified so that the dependence on the post-synaptic activity became monotonic (with saturation) (*Figure 5B*).

We examined how these revised models, in comparison with agents with untrained RNNs that also had non-negative constraints for *x* and *w*, performed in the Pavlovian cue–reward association task examined above (the numbers of RNN units and trials were set to 12 and 1500, respectively). oVRNNbp-rev well developed state values toward reward (*Figure 6A*). oVRNNrf-bio also developed state values to a largely comparable extent (*Figure 6B*). By contrast, the agent with an untrained RNN could not develop such a pattern of state values (*Figure 6C*). This, however, could be because initially set recurrent/feed-forward connections were far from those learned in the online value-RNNs. Therefore, as a more strict control, we conducted simulations of agents with untrained RNN with non-negative *x* and *w*, where in each simulation the recurrent/feed-forward connections were set to be those shuffled from the learned connections in a simulation of oVRNNrf-bio (hereafter we refer to these two types of untrained RNN as 'naive untrained RNN' and 'shuffled untrained RNN'). The model with shuffled untrained RNN developed state values somewhat better than the naive untrained RNN case (*Figure 6D*), but still worse than oVRNNbp-rev and oVRNNrf-bio.

## Systematic simulations and analyses

We varied the number of RNN units (*n*), with the learning rate for value weights normalized by dividing by *n*/12, and compared the performance (mean of squared errors of state values between cue and reward at 1500th trial) of oVRNNbp-rev and oVRNNrf-bio, in comparison with models with naive or shuffled untrained RNN. As shown in the left panel of *Figure 6E*, oVRNNbp-rev and oVRNNrf-bio exhibited largely comparable performance and always outperformed the models with untrained RNN

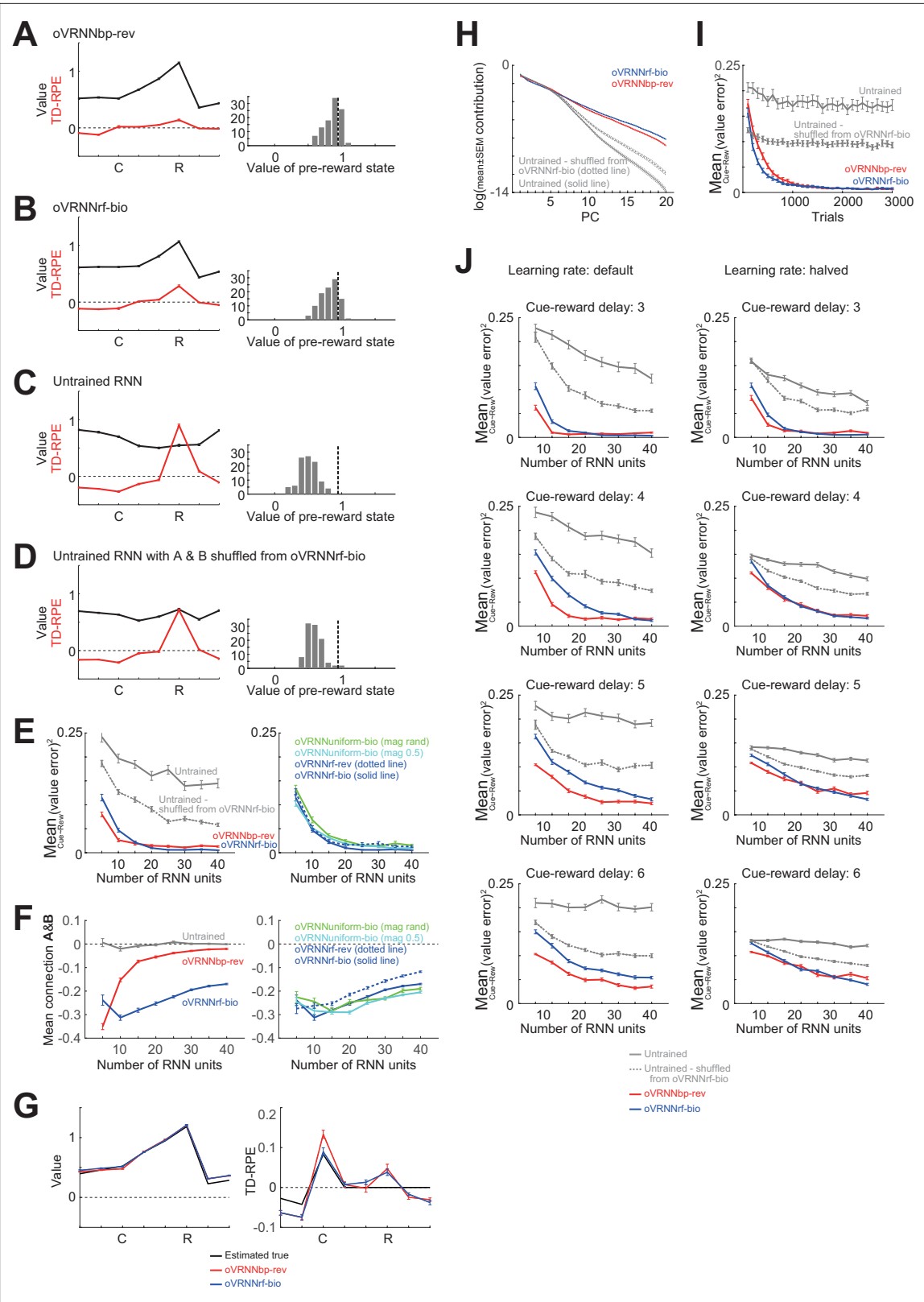

**Figure 6.** Performances of the revised online value-recurrent neural network (RNN) models in the cue–reward association task, in comparison with models with untrained RNN that also had the non-negative constraint. State values (black lines) and TD-RPEs (red lines) at the 1500th trial in oVRNNbp-rev (**A**), oVRNNrf-bio (**B**), agent with naive untrained RNN (i.e., randomly initialized RNN) with *x* and *w* constrained to be non-negative (**C**), and agent with untrained RNN with connections shuffled from those learned in oVRNNrf-bio and also with non-negative *x* and *w* (**D**). The number of RNN units

*Figure 6 continued on next page*

*Figure 6 continued*

was 12 in all the cases. Error bars indicate mean ± SEM across 100 simulations; same applied to the followings unless otherwise mentioned. The right histograms show the across-simulation distribution of the value of the pre-reward state in each model. The vertical black dashed lines in the histograms indicate the true value of the pre-reward state (estimated through simulations). (**E**) *Left*: Mean squared value-error at the 1500th trial in oVRNNbp-rev (red line), oVRNNrf-bio (blue line), agent with naive untrained RNN (gray solid line: partly out of view), and agent with shuffled untrained RNN (gray dotted line) when the number of RNN units (**n**) was varied from 5 to 40. Learning rate for value weights was normalized by dividing by $n/12$ (same applied to the followings). *Right*: Mean squared value-error in oVRNNrf-bio (blue line: same data as in the left panel), oVRNN-bio with random-magnitude uniform feedback (green line), oVRNN-bio with fixed-magnitude (0.5) uniform feedback (light blue line), and oVRNNrf-rev where the update rule of oVRNNrf-bio was changed back to the original one (blue dotted line). (**F**) *Left*: Mean of the elements of the recurrent and feed-forward connections (at 1500th trial) of oVRNNbp-rev (red line), oVRNNrf-bio (blue line), and naive untrained RNN (gray solid line). *Right*: Mean of the elements of the recurrent and feed-forward connections of oVRNNrf-bio (blue line: same data as in the left panel), oVRNN-bio with random-magnitude uniform feedback (green line), oVRNN-bio with fixed-magnitude (0.5) uniform feedback (light blue line), and oVRNNrf-rev (blue dotted line). (**G**) Learned state values (left panel) and TD-RPEs (right panel) in oVRNNbp-rev (red lines) and oVRNNrf-bio (blue lines) in the cases with 40 RNN units, compared to the estimated true values (black lines). (**H**) Log of contribution ratios of the principal components of the time series (for 1500 trials) of RNN activities in each model with 20 RNN units. (**I**) Mean squared value-error in each model with 20 RNN units across trials. (**J**) Mean squared value-error at 3000th trial in each model in the cases where the cue–reward delay was 3, 4, 5, or 6 time steps (top to bottom panels). Left and right panels show the results with default learning rates and halved learning rates, respectively.

($p < 2.5 \times 10^{-12}$ in Wilcoxon rank sum test for oVRNNbp-rev or oVRNNrf-bio vs naive or shuffled untrained for each number of RNN units), although oVRNNbp-rev somewhat outperformed or underperformed oVRNNrf-bio when the number of RNN units was small (≤10 ($p < 0.00029$)) or large (≥25 ($p < 3.7 \times 10^{-6}$)), respectively (*Figure 6G* shows the learned state values and TD-RPEs in oVRNNbp-rev and oVRNNrf-bio in the cases with 40 RNN units, compared to the estimated true values). Remarkably, oVRNNrf-bio generally achieved better performance than both oVRNNbp and oVRNNrf, which did not have the non-negative constraint (Wilcoxon rank sum test, vs oVRNNbp: $p < 7.8 \times 10^{-6}$ for 5 or ≥25 RNN units; vs oVRNNrf: $p < 0.021$ for ≤10 or≥20 RNN units).

The left panel of *Figure 6F* shows the mean of the elements of the recurrent and feed-forward connections at the 1500th trial in the different models. As shown in this figure, these connections (initialized to pseudo standard normal random numbers) were learned to become negative on average, in oVRNNbp-rev and oVRNNrf-bio. This learned negative-dominance (inhibition-dominance) could possibly be related, for example, through prevention of excessive activity, to the good performance of oVRNNrf-bio and also the better performance of the shuffled untrained RNN than the naive untrained RNN. The green and light blue lines in the right panels of *Figure 6E, F* show the results for special cases where the random feedback in oVRNNrf-bio was fixed to the direction of $(1, 1,...,1)^\top$ (i.e., uniform feedback) with a random non-negative magnitude (green line) or a fixed magnitude of 0.5 (light blue line). The performance of these special cases, especially the former (with random magnitude), was somewhat worse than that of oVRNNrf-bio, but still better than that of the models with untrained RNN. The blue dotted lines in the right panels of *Figure 6E, F* show the results where the modified update rule of oVRNNrf-bio was changed back to the original rule with non-monotonic dependence on the post-synaptic activity (*Figure 5A*). The performance of this model was somewhat worse than oVRNNrf-bio, indicating that the biologically motivated modification of the update rule in fact improved the performance.

*Figure 6H* shows contribution ratios of PCs of the time series of RNN activities in each model with 20 RNN units. Compared with the cases with naive/shuffled untrained RNN, in oVRNNbp-rev and oVRNNrf-bio, later components had relatively high contributions (PC5–20, $p < 1.4 \times 10^{-6}$ (*t*-test vs naive) or <0.014 (vs shuffled) in oVRNNbp-rev; PC6–20, $p < 2.0 \times 10^{-7}$ (vs naive) or PC7–20, $p < 5.9 \times 10^{-14}$ (vs shuffled) in oVRNNrf-bio), explaining their superior value-learning performance. *Figure 6I* shows how learning proceeded across trials in the models with 20 RNN units. While oVRNNbp-rev and oVRNNrf-bio eventually reached a comparable level of errors, oVRNNrf-bio outperformed oVRNNbp-rev in early trials (at 200, 300, 400, or 500 trials; $p < 0.049$ in Wilcoxon rank sum test for each). This is presumably because the value weights did not develop well in early trials, and so the backprop-type feedback, which was the same as the value weights, did not work well, while the non-negative fixed random feedback worked finely from the beginning. *Figure 6J* shows the cases with longer cue–reward delays, with default or halved learning rates. As the delay increased, the mean squared error of state values (at 3000th trial) increased, but the relative superiority of oVRNNbp-rev and oVRNNrf-bio over the models with untrained RNN remained to hold, except for a few cases with 5 RNN units (5

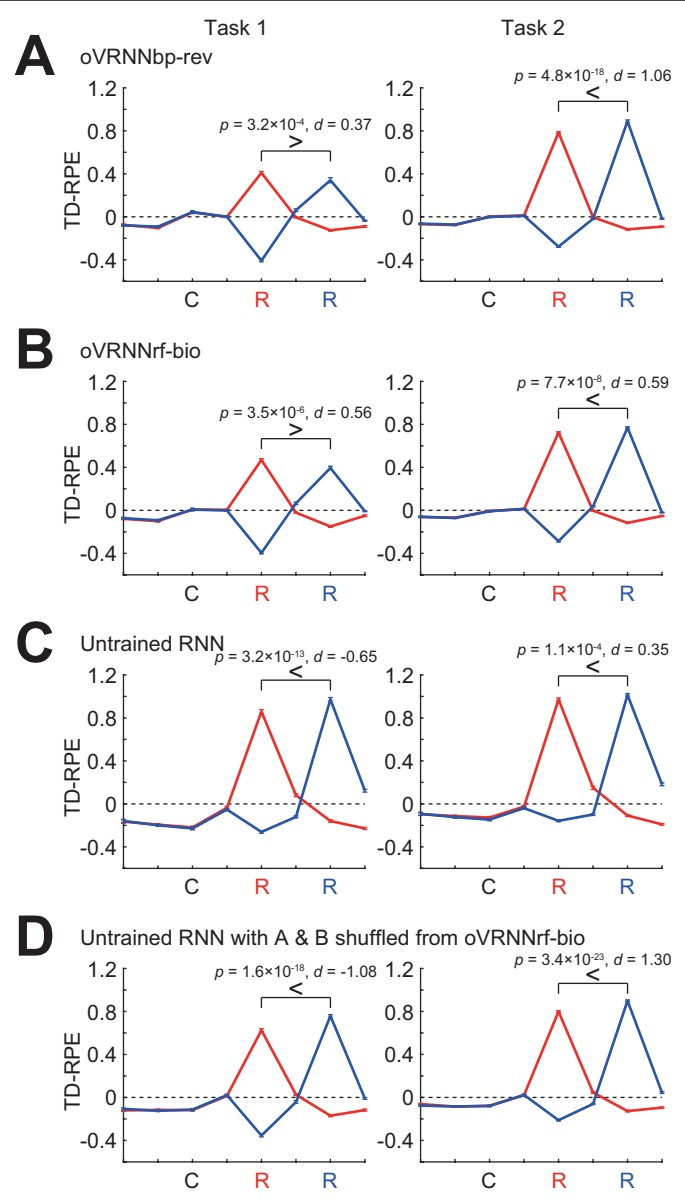

**Figure 7.** Performances of the revised online value-recurrent neural network (RNN) models with further biological constraints in the two tasks having probabilistic structures, in comparison with models with untrained RNN. TD-RPEs at the latest trial within 2000 trials in which reward was given at the early timing (red lines) or the late timing (blue lines) in task 1 (left) and task 2 (right), averaged across 100 simulations (error bars indicating ± SEM across simulations), are shown for the four types of agent: (**A**) oVRNNbp-rev; (**B**) oVRNNrf-bio; (**C**) agent with naive untrained RNN; (**D**) agent with untrained RNN with connections shuffled from those learnt in oVRNNrf-bio. The number of RNN units was 20 for all the cases.

delay oVRNNrf-bio vs shuffled with default learning rate, 6 delay oVRNNrf-bio vs naive or shuffled with halved learning rate) ($p < 0.047$ in Wilcoxon rank sum test for oVRNNbp-rev or oVRNNrf-bio vs naive or shuffled untrained for each number of RNN units for each delay). We further examined how the revised online value-RNN models performed in the two tasks with probabilistic structures examined above. The models with 12 RNN units appeared not able to produce the expected different patterns of TD-RPEs in the two tasks (TD-RPE at early reward > TD RPE at late reward in task 1 and opposite pattern in task 2), and we increased the number of RNN units to 20. Then, both oVRNNbp-rev and oVRNNrf-bio produced such TD-RPE patterns (**Figure 7A, B**), whereas the models with untrained

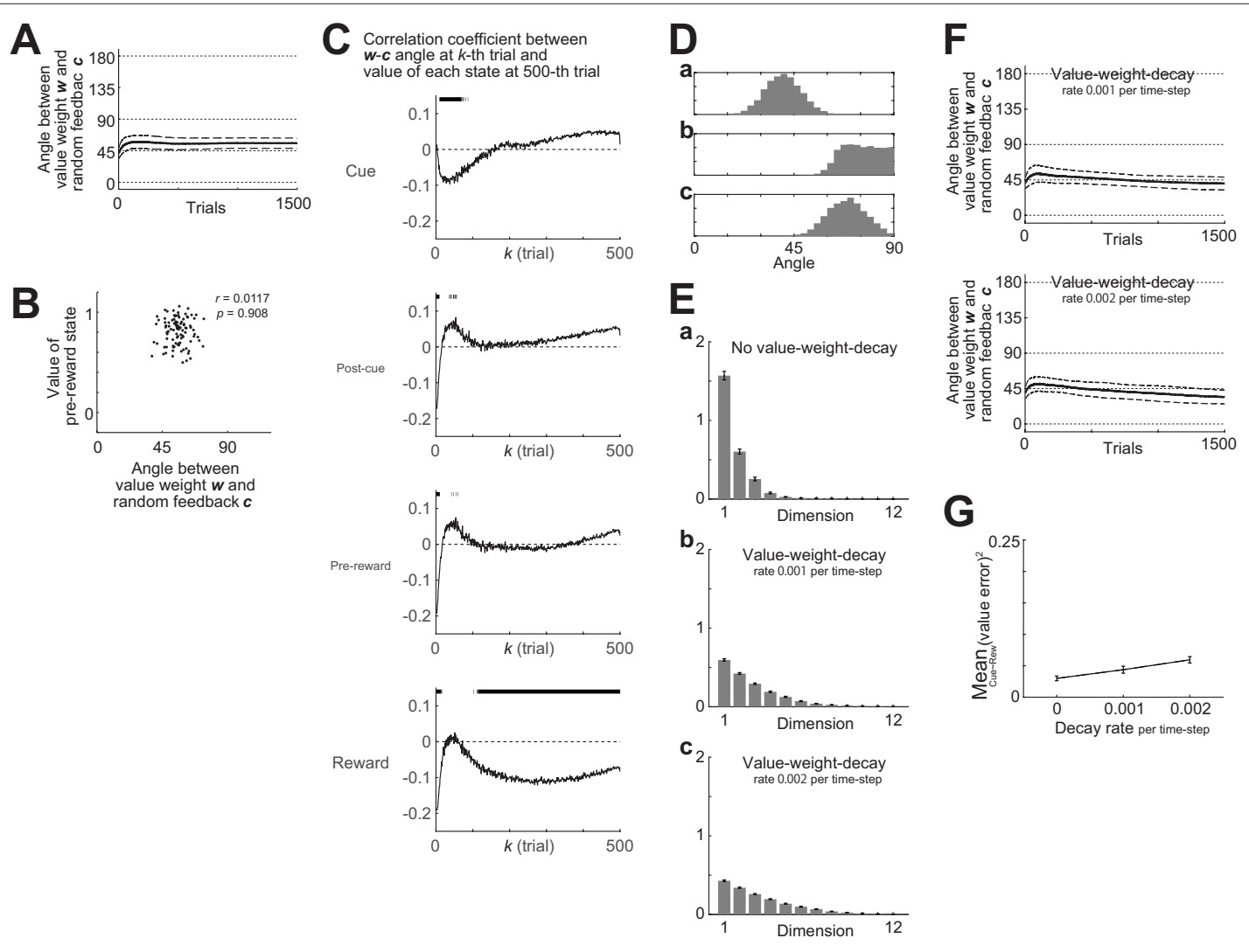

**Figure 8.** Loose alignment of the value weights (*w*) and the random feedback (*c*) in oVRNNrf-bio (with 12 recurrent neural network [RNN] units). (**A**) Over-trial changes in the angle between the value weights *w* and the fixed random feedback *c*. The solid line and the dashed lines indicate the mean ± SD across 100 simulations, respectively. (**B**) No correlation between the *w*–*c* angle (horizontal axis) and the value of the pre-reward state (vertical axis) at 1500th trial (*r* = 0.0117, p = 0.908). The dots indicate the results of individual simulations. (**C**) Correlation between the *w*–*c* angle at *k*th trial (horizontal axis) and the value of the cue, post-cue, pre-reward, or reward state (top-bottom panels) at 500th trial across 1000 simulations. The solid lines indicate the correlation coefficient, and the short vertical bars at the top of each panel indicate the cases in which p-value was less than 0.05. (**D**) Distribution of the angle between two 12-dimensional vectors when the elements of both vectors were drawn from [0 1] uniform pseudo-random numbers (**a**) or when one of the vectors was replaced with [1 0 0... 0] (i.e., on the edge of the non-negative quadrant) (**b**) or [1 1 0... 0] (i.e., on the boundary of the non-negative quadrant) (**c**). (**E**) Across-simulations histograms of elements of *w* in oVRNNrf-bio with 12 RNN units ordered from the largest to smallest ones after 1500 trials when there was no value-weight-decay (**a**) or there was value-weight-decay with decay rate (per time step) of 0.001 (**b**) or 0.002 (**c**). The error bars indicate the mean ± SEM across 100 simulations. (**F**) Over-trial changes in the angle between the value weights *w* and the fixed random feedback *c* when there was value-weight decay with decay rate (per time step) of 0.001 (top panel) or 0.002 (bottom panel). Notations are the same as those in (**A**). (**G**) Mean squared value-error at the 1500th trial in oVRNNrf-bio with 12 RNN units with the rate of value-weight decay varied (horizontal axis). The error bars indicate the mean ± SEM across 100 simulations.

RNN could not (*Figure 7C, D*). This indicates that the online value-RNN with random feedback and further biological constraints could learn the differential characteristics of the tasks.

### Loose alignment and FA
Coming back to the original cue–reward association task, we examined how the angle between the value weights (*w*) and the random feedback (*c*) changed across trials in oVRNNrf-bio with 12 RNN

units. As shown in *Figure 8A*, the angle was on average smaller than 90°, which was the chance-level angle in the case without non-negative constraint, from the beginning, while there was no further alignment over trials. This could be understood as follows. Because both the value weights (*w*) and the random feedback (*c*) were now constrained to be non-negative, these two vectors were ensured to be in a relatively close angle (i.e., in the same quadrant) from the beginning. By virtue of this loose alignment, the random feedback could act similarly to backprop-type transported-weight feedback, even without further alignment.

We examined if the angle between the value weights (*w*) and the random feedback (*c*) at the 1500th trial was associated with the developed value of pre-reward state across simulations, but found no association (*r* = 0.0117, p = 0.908) (*Figure 8B*). We then examined if the *w–c* angle at earlier trials (2nd to 500th trials) was associated with the developed values at 500th trial, with the number of simulations increased to 1000 so that small correlation could be detected. We found that the *w–c* angle at initial trials (2nd to around 10th trials) was negatively correlated with the developed values of the reward state and preceding states at 500th trial (*Figure 8C*). As for the reward state, negative correlation at around 100th to 300th trial was also observed. These results suggest that better alignment of *w* and *c* at initial and early timings was associated with better development of state values, in line with the conjecture that loose alignment of *w* and *c* coming from the non-negative constraint supported learning. It should be noted, however, that there were cases where positive (although small) correlation was observed. Its exact reason is not sure, but it could be related to the fact that the largeness of developed values or the speed of value development does not necessarily mean good learning.

As mentioned above, while the angle between *w* and *c* was on average smaller than 90° from the beginning, there was no further alignment over trials. This seemed mysterious because the mechanism for FA that we derived for the models without non-negative constraint was expected to work also for the models with non-negative constraint. As a possible reason for the non-occurrence of FA, we guessed that one or a few element(s) of *w* grew prominently during learning, and so *w* became close to an edge or boundary of the non-negative quadrant and thereby the angle between *w* and other vector became generally large (as illustrated in *Figure 8D*). *Figure 8Ea* shows the mean ± SEM of the elements of *w* ordered from the largest to smallest ones after 1500 trials. As conjectured above, a few elements indeed grew prominently.

We considered that if a slight decay (forgetting) of value weights (cf., *Morita and Kato, 2014*; *Kato and Morita, 2016*; *Kato and Morita, 2025*) was assumed, such a prominent growth of a few elements of *w* may be mitigated and alignment of *w* to *c*, beyond the initial loose alignment because of the non-negative constraint, may occur. These conjectures were indeed confirmed by simulations (*Figure 8Eb, c, F*). The mean squared value-error slightly increased when the value-weight decay was assumed (*Figure 8G*); however, presumably reflecting a decrease in developed values and a deterioration of learning because of the decay.

## Models with excitatory and inhibitory units

As mentioned above, in oVRNNbp-rev and oVRNNrf-bio, the connection weights in/onto the RNN could be both positive and negative, against Dale's law. Recent studies started to examine neural networks incorporating Dale's law (*Cornford et al., 2021*; *Li et al., 2023*) or other connectivity features (*Mastrogiuseppe and Ostojic, 2018*). So we examined extended models, named oVRNNbp-rev-ei and oVRNNrf-bio-ei, which incorporated excitatory E-units, modeling pyramidal cells, and inhibitory I-units, modeling fast-spiking (FS) cells (*Figure 9A*). Cortical excitation can operate slowly due to slow synaptic dynamics (*Mongillo et al., 2008*; *Morishima et al., 2011*) (see the description about the time step in the Methods for details). In contrast, inhibition from FS cells to pyramidal cells may operate more quickly, since it was shown (*Morita et al., 2008*) that observed phases of regular-spiking (RS, putatively pyramidal) cells' and FS cells' spikes (*Hasenstaub et al., 2005*) could be explained by fast FS → RS inhibition and temporally distributed recurrent excitation.

Given these, we assumed that excitation from E-units to E- and I-units took one time step whereas I → E inhibition operated within a time step, and also that each E-unit received inputs from all the E-units and a particular I-unit (although this assumption could be supported by the abovementioned suggestions, its validity remains largely open). Chemical and electrical connections between FS cells exist and are suggested to serve for synchronization or oscillation (*Wang, 2010*; *Buzsáki and Wang, 2012*), but we omitted I → I connections because our models did not describe fast spike dynamics.

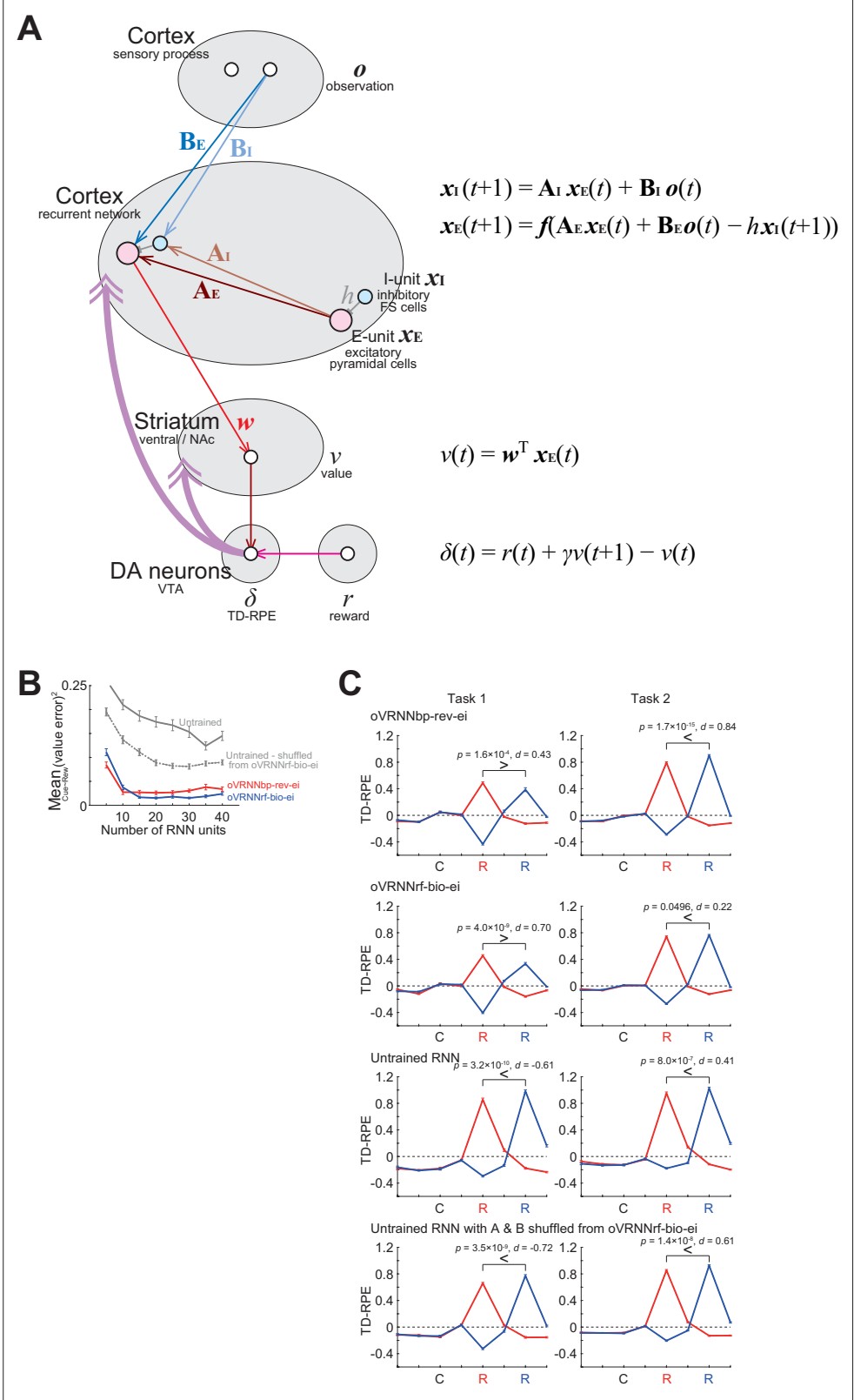

**Figure 9.** oVRNNbp-rev-ei and oVRNNrf-bio-ei models incorporating excitatory E-units and inhibitory I-units. (**A**) Schematic illustration of the models' architecture. For ease of viewing, only limited parts of units and connections are drawn. (**B**) Mean squared value-error at the 1500th trial in the cue–reward association task in oVRNNbp-rev-ei (red line), oVRNNrf-bio-ei (blue line), and E-/I-units-incorporated models with naive untrained recurrent neural

*Figure 9 continued on next page*

*Figure 9 continued*

network (RNN) (i.e., randomly initialized RNN) (gray solid line) or untrained RNN with connections shuffled from those learned in oVRNNrf-bio-ei (gray dotted line). (**C**) Patterns of TD-RPE in the tasks with probabilistic structures generated in the four models with E-/I-units. Simulation conditions and notations are the same as those in *Figure 7*.

Since FS cells can fire at high frequencies, we assumed that the activation function for I-units was not saturating but linear. Lastly, we did not assume plasticity for connections from/to I-units. The connection weights onto the E- and I-units from the observation units and E-units were non-negatively initialized, specifically, initialized to pseudo normal random numbers with mean = 3 and SD = 1 and rectified to 0 when becoming negative.

We examined how these extended models behaved in the Pavlovian task and the probabilistic tasks. As shown in *Figure 9B*, oVRNNbp-rev-ei and oVRNNrf-bio-ei learned the state values in the Pavlovian task much more accurately than the models with naive untrained RNN or untrained RNN whose connections from the observation and E-units to E- and I-units were shuffled from oVRNNrf-bio-ei ($p < 5.2 \times 10^{-12}$ in Wilcoxon rank sum test for oVRNNbp-rev-ei or oVRNNrf-bio-ei vs naive or shuffled untrained for each number of RNN units). oVRNNbp-rev-ei somewhat outperformed or underperformed oVRNNrf-bio-ei when the number of RNN units was small ($\leq10$ ($p < 0.0091$)) or relatively large (15–35 ($p < 0.027$)), respectively. Also, as shown in *Figure 9C*, oVRNNbp-rev-ei and oVRNNrf-bio-ei with 20 E-units and 20 I-units generated the expected different patterns of TD-RPEs in the two tasks (TD-RPE at early reward > TD-RPE at late reward in task 1 and opposite pattern in task 2), although the effect size for task 2 in oVRNNrf-bio-ei was small, while the models with untrained RNN did not.

As such, the extended models with E- and I-units showed largely similar behaviors to those of the original oVRNNbp-rev and oVRNNrf-bio with mixed positive and negative RNN weights. This is actually reasonable, because combining the update equations for I- and E-units in the extended models (top two equations in *Figure 9A*) results in an equation largely similar to the update equation for RNN units in the original models (top equation in *Figure 1*). In other words, the original models with mixed positive and negative RNN weights could be regarded as a simplified description of the models with E- and I-units under the abovementioned assumptions. Therefore, for simplicity, we will return to mixed positive and negative RNN weights in the following.

## Task with distractor cue

So far, we have examined situations where there existed a reward and a cue associated with the reward. However, in real environments, it is likely that there exist both reward-associated and non-associated (distractor) cues, and the agent does not initially know which cue is associated with reward and which is not. Learning cue–reward association in such distractor-existing environments is generally not easy for biologically constrained models, and it has been addressed by only a few previous works (*Cone et al., 2024*). We examined whether our biologically constrained oVRNNrf-bio, as well as oVRNNbp-rev, could learn the cue–reward association under the presence of distractor cue. We considered a simple case where there existed a distractor cue, which was presented to the agent with a certain probability at every time step, between cue–reward duration or reward–cue duration (i.e., ITI) or simultaneously with cue or reward. As for the agent's models, we assumed that the observation inputs had an additional element (dimension), which was set to 1 when the distractor was presented and 0 when not (*Figure 10A*).

We examined how oVRNNbp-rev, oVRNNrf-bio, and the models with untrained RNN behaved in the modified Pavlovian task with a distractor cue, which was presented with probability 0, 0.1, 0.2, or 0.3 at every time step (*Figure 10B–E*, left panels). As a result, even when there was such a distractor cue, oVRNNbp-rev and oVRNNrf-bio could still learn the state values better than the models with naive or shuffled untrained RNN ($p < 1.7 \times 10^{-10}$ in Wilcoxon rank sum test for oVRNNbp-rev or VRNNrf-bio vs naive or shuffled untrained for each number of RNN units for each level of distractor probability) (*Figure 10C–E*, middle panels), although the accuracy moderately decreased compared with the case without distractor cue (*Figure 10B*, middle panel). These results suggest robustness of the learning ability of oVRNNrf-bio against distractor in realistic situations. We further examined how the models with E- and I-units behaved in the task with a distractor cue and confirmed that even in the

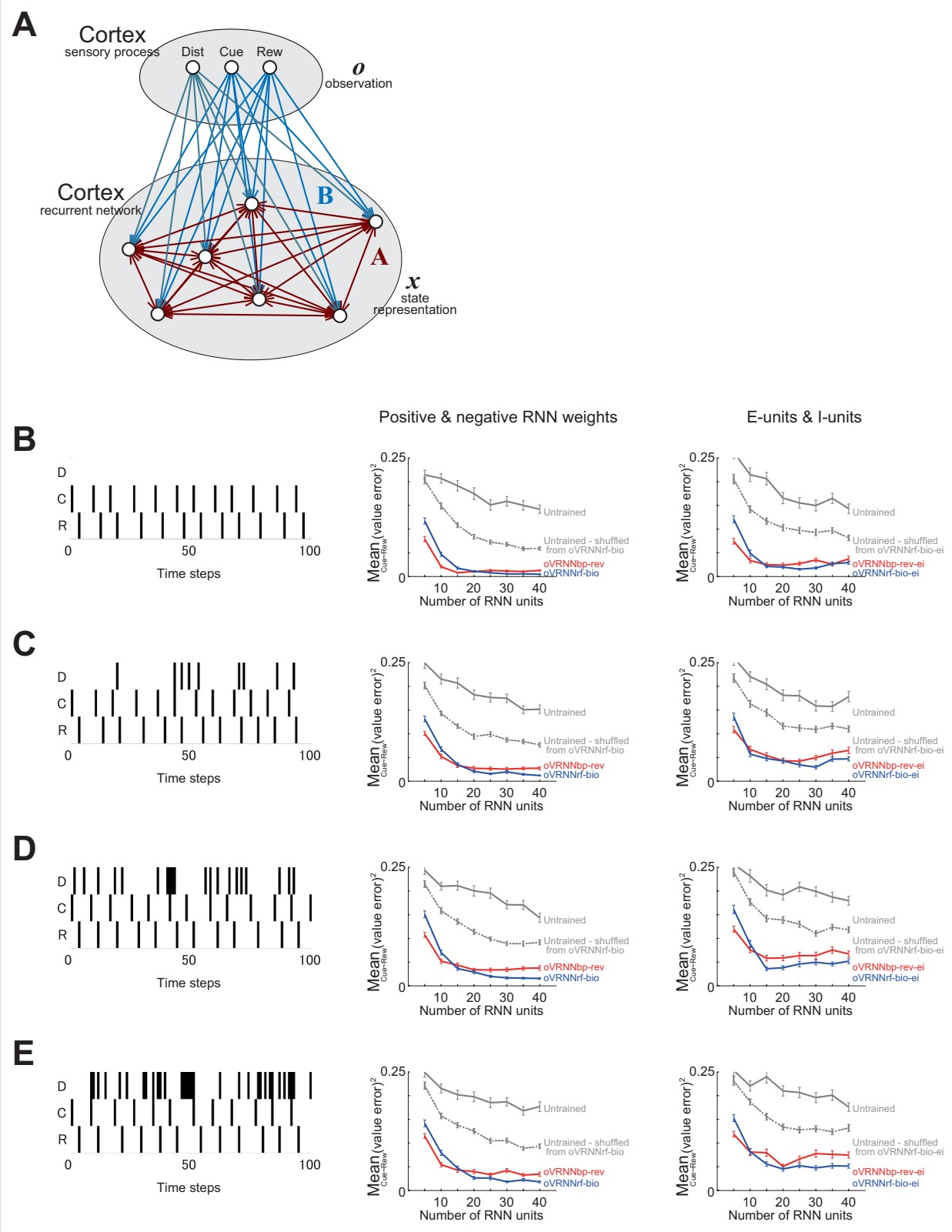

**Figure 10.** Cue–reward association task with distractor cue. (**A**) Modification of oVRNNbp-rev and oVRNNrf-bio to incorporate possible existence of distractor cue. The observation units *o* had an additional element (leftmost circle labeled as 'Dist'), which was 1 at the time steps where distractor cue was present and 0 otherwise. Results of the cases where the probability of the presence of distractor cue at every time step was 0 (**B**), 0.1 (**C**), 0.2 (**D**), and 0.3 (**E**). *Left panels*: Examples of the presence of distractor ('D'), reward-associated cue ('C'), and reward ('R') over 100 time steps. *Middle panels*: Mean

*Figure 10 continued on next page*

*Figure 10 continued*

squared value-error at the 1500th trial in oVRNNbp-rev (red line), oVRNNrf-bio (blue line), and the models with naive or shuffled untrained recurrent neural network (RNN) (gray solid or dotted line). *Right panels*: Results for the models with E- and I-units (oVRNNbp-rev-ei: red line, oVRNNrf-bio-ei: blue line, models with naive or shuffled untrained RNN: gray solid or dotted line), which were modified to incorporate possible existence of distractor cue in the same manner as in (**A**).

presence of a distractor cue, oVRNNbp-rev-ei and oVRNNrf-bio-ei could learn the state values better than the models with naive or shuffled untrained RNN (p < 3.2 × $10^{-8}$ in Wilcoxon rank sum test for oVRNNbp-rev-ei or oVRNNrf-bio-ei vs naive or shuffled untrained for each number of RNN units for each level of distractor probability) (*Figure 10B–E*, right panels).

## Incorporation of action selection

Ultimate purpose of animals and RL agents is to optimize their policy, that is, probability of action selection at each state to maximize rewards. Therefore, we examined if our models could be extended to incorporate action selection. In reference to the proposals that algorithms akin to the actor-critic method may be implemented in the brain (*Bellec et al., 2020*; *Sutton and Barto, 1998*; *Takahashi et al., 2008*), we considered extended models oVRNNbp-rev-ac and oVRNNrf-bio-ac, which incorporated an actor-critic architecture (*Figure 11A*). Specifically, each RNN unit was assumed to connect to not only the state-value ($v$)-representing unit in the ventral striatum but also the action-value ($q_k$)-representing units in the dorsal striatum. Their non-negative weights ($u_{kj}$) represented (as a vector) action preferences, which slightly decayed with time so as not to unboundedly increase.

It has been suggested that action is selected through competition of neural populations in the striatum–thalamus–cortex(-striatum) circuit, which represent or receive action values, in the presence of noise (*Wang, 2002*; *Lo and Wang, 2006*; *Hunt et al., 2012*). However, because our models did not describe fast neural dynamics (see the description about the time step in the Methods), we assumed a soft-max function for action selection, i.e., assumed that an action was selected according to a soft-max probability determined by the difference in the action values when there were two action candidates. We then assumed that there is a cortical region, which contains action-representing neural populations, implemented as 'action units' in the model (*Figure 11A*). Action unit was assumed to become active (i.e., = 1) when the corresponding action was selected and inactive (i.e., = 0) otherwise, and these action units were assumed to send inputs to the RNN, similarly to the observation units informing the presence of cue and reward.

We examined how these models (oVRNNbp-rev-ac and oVRNNrf-bio-ac) and control models with naive or shuffled untrained RNNs behaved in two-alternative choice tasks. In the first choice task (*Figure 11Ba*), taking action 1 led to a large (size 2) reward two time steps later whereas taking action 2 led to a small (size 1) reward two time steps later. oVRNNbp-rev-ac and oVRNNrf-bio-ac, after training, successfully selected the better action, action 1, in most cases, whereas the models with untrained RNN did not develop strong tendency to select action 1 (*Figure 11Bb, c*). Next, in the second choice task (*Figure 11Ca*), taking action 1 led to a large (size 2) reward two time steps later whereas taking action 2 led to a small (size 1) reward one time step later. So this task imposed inter-temporal choice between delayed large reward and sooner small reward. oVRNNbp-rev-ac, after training, tended to select action 1, which was better than action 2 even when presumed temporal discounting (0.8 per time step) was taken into account (*Figure 11Cb, c*). oVRNNrf-bio-ac also tended to select action 1 when the number of RNN units was not small. On the other hand, the models with untrained RNNs tended to select small rewards sooner. These results suggest that oVRNNrf-bio-ac, as well as oVRNNbp-rev-ac, could learn to select advantageous action to a certain extent, even in the case of inter-temporal choice.

## Discussion

We have shown that state representation and value can be learned online in the RNN and its readout by using random feedback instead of biologically unavailable downstream weights. This was achieved through FA, and we have presented an intuitive understanding of its mechanism. We have further shown that the non-negative constraint realizes loose alignment of the forward weights and feedback from the beginning, which appeared to support learning.

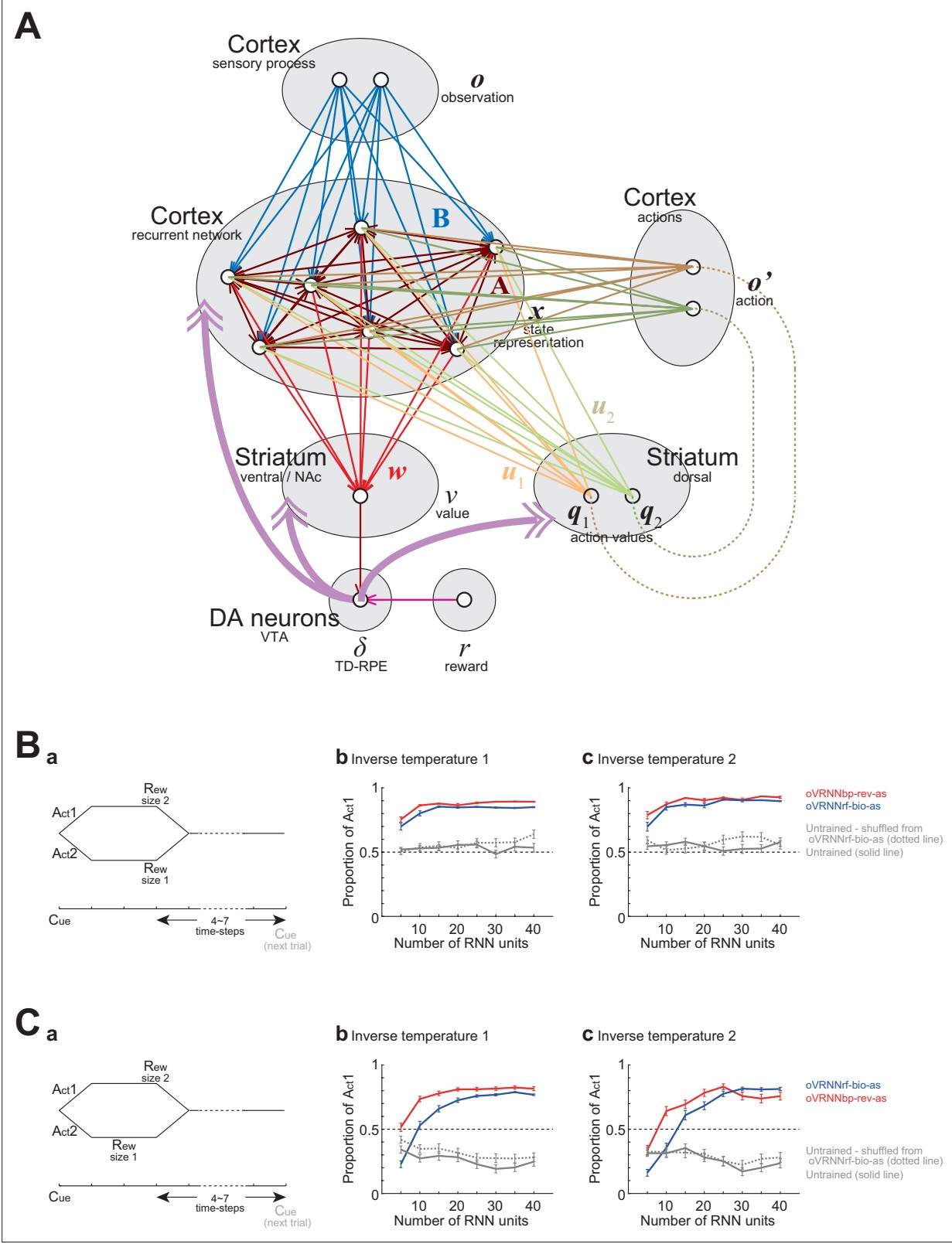

**Figure 11.** Incorporation of action selection. (**A**) Schematic illustration of the models incorporating an actor-critic architecture. (**B**) Two-alternative choice task. (**a**) Task diagram. (**b, c**) Proportion of Action 1 selection in 2901–3000th trials in oVRNNbp-rev-as (red line), oVRNNrf-bio-as (blue line), and the models with naive untrained recurrent neural network (RNN) (i.e., randomly initialized RNN) (gray solid line) or untrained RNN with connections shuffled from those learnt in VRNNrf-bio-as (gray dotted line). Error bars indicate the mean ± SEM over 100 simulations. The inverse temperature was set to 1 (**b**) or 2 (**c**). (**C**) Inter-temporal choice task. The notations are the same as those in (**B**).

## Roles of DA

Midbrain DA neurons project to both striatum and cortex, including the prefrontal cortex (*Williams and Goldman-Rakic, 1998*) and the hippocampus (*Broussard et al., 2016*). As for striatal DA, DA-dependent cortico-striatal plasticity is considered to implement TD-RPE-based value update (*Reynolds et al., 2001*; *Samejima et al., 2005*). By contrast, while cortical DA has been implicated in working memory (*Brozoski et al., 1979*; *Sawaguchi and Goldman-Rakic, 1991*; *Durstewitz et al., 2000*; *Brunel and Wang, 2001*), decision making (*Floresco and Magyar, 2006*), and aversive memory (*Tsetsenis et al., 2021*), its role as TD-RPE remains unclear, despite findings suggesting that cortical DA does encode (TD-)RPE (*O'Doherty et al., 2003*; *Takahashi et al., 2011*; *Starkweather et al., 2018*; *Takahashi et al., 2023*) and modulate plasticity (*Otani et al., 2003*; *Sayegh et al., 2024*). Learnability of our biologically constrained online value-RNN suggests that TD-RPE-encoding cortical DA modulates plasticity of RNN so that appropriate state representation can be learned.

Many studies reported heterogeneities of DA signals, which may come from encoding prediction errors other than RPE or feature-specific components of RPE (*Lee et al., 2024b*). Referring to a result (*Avvisati et al., 2024*) indicating DA's encoding of non-reward PEs and the fact that DA neurons receive inputs from the cerebellum (*Watabe-Uchida et al., 2012*; *Carta et al., 2019*), which presumably implements supervised learning (*Marr, 1969*), a recent work (*Wärnberg and Kumar, 2023*) proposed that DA encodes vector-valued errors used for supervised learning of actions in continuous space. In contrast, we assumed DA's encoding of scalar TD-RPE, which can be consistent with the heterogeneity due to encoding of feature-specific RPE components (*Lee et al., 2024b*). The previous model (*Wärnberg and Kumar, 2023*) and our model can coexist, with different DA neuronal populations encoding different types of errors, or a single DA neuron switching its encoding depending on context/inputs.

VTA DA neurons also project to the basolateral amygdala (BLA) (*Beier et al., 2015*), and DA also regulates plasticity there (*Li and Rainnie, 2014*). Moreover, VTA → BLA DA entailed properties of TD-RPE, although increased also upon aversive event and was not itself reinforcing but crucial for the formation of environmental model (*Sias et al., 2024*). BLA has recurrent connections (*Headley et al., 2021*), projects to the striatum (*Britt et al., 2012*; *Lee et al., 2024a*), and engages in abstract context representation (*Saez et al., 2015*). Thus, given that goal-directed-like behavior could be achieved through sophisticated state representation (*Russek et al., 2017*; *Stachenfeld et al., 2017*; *Qian et al., 2025*), it could potentially be learned by value-RNN-like mechanism in the BLA. Whether such sophisticated representation can be learned, however, remains open, and it might require multi-dimensional error (*Stalnaker et al., 2019*) beyond TD-RPE.

There are many DA-related mechanisms that were not incorporated into our models, including the distinctions of D1-direct and D2-indirect pathways (*Gerfen and Surmeier, 2011*; *Collins and Frank, 2014*; *Mikhael and Bogacz, 2016*; *Morita and Kawaguchi, 2018*; *Lowet et al., 2025*) and cortical projections to them (*Wall et al., 2013*; *Morita, 2014*; *Hooks et al., 2018*; *Morita et al., 2019*), as well as mechanisms underlying TD-RPE encoding (cf., *Morita and Kawaguchi, 2018*; *Tian et al., 2016*) or learning for it (cf., *Cone et al., 2024*). Future studies are expected to incorporate these.

## Predictions and implications of our models

oVRNNrf predicts that the feedback vector $c$ and the value-weight vector $w$ become gradually aligned, while oVRNNrf-bio predicts that $c$ and $w$ are loosely aligned from the beginning. Element of $c$ could be measured as the magnitude of pyramidal cell's response to DA stimulation. The element of $w$ corresponding to a given pyramidal cell could be measured, if striatal neuron that receives input from that pyramidal cell can be identified (although technically demanding), as the magnitude of response of the striatal neuron to activation of the pyramidal cell. Then, the abovementioned predictions could be tested by (1) identifying cortical, striatal, and VTA regions that are connected, (2) identifying pairs of cortical pyramidal cells and striatal neurons that are connected, (3) measuring the responses of identified pyramidal cells to DA stimulation, as well as the responses of identified striatal neurons to activation of the connected pyramidal cells, and (4) testing whether DA → pyramidal responses and pyramidal → striatal responses are associated across pyramidal cells, and whether such associations develop through learning.

Testing this prediction, however, would be technically quite demanding, as mentioned above. An alternative way of testing our model is to manipulate the cortical DA feedback and see if it will cause

(re-)alignment of value weights (i.e., cortical striatal strengths). Specifically, our model predicts that if DA projection to a particular cortical locus is silenced, the effect of the activity of that locus on the value-encoding striatal activity will become diminished.

We have shown that oVRNNrf and oVRNNrf-bio could work even when the random feedback was uniform, that is, fixed to the direction of $(1, 1,..., 1)^T$, although the performance was somewhat worse. This is reasonable because uniform feedback can still encode scalar TD-RPE that drives our models, in contrast to a previous study (*Wärnberg and Kumar, 2023*), which considered DA's encoding of vector error and thus regarded uniform feedback as a negative control. If oVRNNrf/oVRNNrf-bio-like mechanism indeed operates in the brain and the feedback is near uniform, alignment of the value weights *w* to near (1, 1,..., 1) is expected to occur. This means that states are (learned to be) represented in such a way that simple summation of cortical neuronal activity approximates value, thereby potentially explaining why value is often correlated with regional activation (fMRI BOLD signal) of cortical regions (*Levy and Glimcher, 2012*). Notably, uniform feedback coupled with positive forward weights was shown to be effective also in supervised learning of one-dimensional output in feed-forward networks (*Konishi et al., 2023*), and we guess that loose alignment may underlie it.

## On the RNN unit

In our oVRNNbp without non-negative constraint, as the number of RNN units increased, the squared error initially decreased but then increased (*Figure 2J*) (while intriguingly it was not the case for the models with non-negative constraint (*Figure 6E*)). In contrast, in the original value-RNN (*Qian et al., 2025*; *Hennig et al., 2023*), the ability to develop belief-state-like representation was reported to improve as the number of RNN units increased to 100 or 50. There are at least two possible reasons for this difference, other than the difference in the performance measures. The first one is the difference in the update rules. As mentioned earlier, the original value-RNN used BPTT (*Rumelhart et al., 1986b*) whereas our oVRNNbp used an online learning rule, which only considered the influence of the recurrent weights at the previous time step.

The second one is a difference in the RNN unit. Specifically, the original value-RNN used the Gated Recurrent Unit (GRU) cell (*Cho et al., 2014*) whereas we used a simple sigmoidal function. RNN with simple nonlinear unit is known to have the vanishing gradient problem (*Hochreiter, 1998*), which could be alleviated by using memorable/gated RNN such as the Long Short-Term Memory (LSTM) unit (*Hochreiter and Schmidhuber, 1997*) or the GRU cell (*Cho et al., 2014*). We used the simple sigmoidal unit because the biological plausibility of the GRU cell appeared elusive. However, a gated unit similar to the LSTM unit has actually been proposed to be implemented in cortical microcircuits (*Costa et al., 2017*), and incorporation of such biologically plausible gated unit into our online value-RNN would be a hopeful direction.

From a bottom-up viewpoint, our RNN unit did not incorporate spiking (*Payeur et al., 2021*; *Gjorgjieva et al., 2011*; *Shouval et al., 2010*) nor nonlinear dendritic computations (*Pagkalos et al., 2024*; *Poirazi et al., 2003*; *Morita, 2008*). Recent studies suggest that dendritic mechanisms (*Guerguiev et al., 2017*; *Sacramento et al., 2018*), possibly in combination with burst-dependent plasticity (*Payeur et al., 2021*; *Greedy et al., 2022*), can realize credit assignment without backprop in supervised and unsupervised learning (*Körding and König, 2001*; *Illing et al., 2021*). Also, a recent model of hippocampus *Cone and Clopath, 2024* has shown that a network of multi-compartment units could learn complex representations. Having dendritic mechanisms is different from just increasing the number of neural-network layers because of their own specific features/constraints, and it was argued (*Pagkalos et al., 2024*) that adding such biological constraints enables learning in deep neural networks. Therefore, incorporation of biological details into RNN unit in our models would be hopeful also from the bottom-up viewpoint.

## Comparison to other algorithms

As an alternative to backprop in hierarchical network, aside from FA (*Lillicrap et al., 2016*), Associative Reward–Penalty ($A_{R–P}$) algorithm has been proposed (*Barto and Jordan, 1987*; *Mazzoni et al., 1991a*; *Mazzoni et al., 1991b*). In $A_{R–P}$, the hidden units behave stochastically, allowing the gradient to be estimated via stochastic sampling. Recent work by *Max et al., 2024* has proposed Phaseless Alignment Learning, in which high-frequency noise-induced learning of feedback projections proceeds simultaneously with learning of forward projections using the feedback in a lower frequency. Noise-induced

learning of the weights on readout neurons from untrained RNN by reward-modulated Hebbian plasticity has also been demonstrated (*Hoerzer et al., 2014*). Such noise- or perturbation-based (*Lillicrap et al., 2020*) mechanisms are biologically plausible because neurons and neural networks can exhibit noisy or chaotic behavior (*Faisal et al., 2008*; *Aihara and Matsumoto, 1986*; *van Vreeswijk and Sompolinsky, 1996*), and might improve the performance of value-RNN if implemented.

Regarding the learning of RNN, 'e-prop' (*Bellec et al., 2020*) was proposed as a locally learnable online approximation of BPTT (*Rumelhart et al., 1986b*), which was used in the original value-RNN (*Hennig et al., 2023*). In e-prop, neuron-specific learning signal is combined with weight-specific locally updatable 'eligibility trace'. Reward-based e-prop was also shown to work (*Bellec et al., 2020*), both in a setup not introducing TD-RPE with symmetric or random feedback (their Figure S5) and in another setup introducing TD-RPE with symmetric feedback (their Figures 4 and 5). Compared to these, our models differ in multiple ways.

First, we have shown that alignment to random feedback occurs in the models driven by TD-RPE. Second, our models do not have 'eligibility trace' (nor memorable/gated unit, different from the original value-RNN *Hennig et al., 2023*; see Appendix 1.1 for possible incorporation of eligibility trace into our models), but could still solve temporal credit assignment to a certain extent because TD learning is by itself a solution for it (notably, recent work showed that combination of TD(0) and model-based RL well explained rat's choice and DA patterns; *Krausz et al., 2023*). However, as mentioned before, a single time step in our models was assumed to correspond to hundreds of milliseconds, incorporating slow synaptic dynamics, whereas e-prop is an algorithm for spiking neuron models with a much finer time scale. From this aspect, our models could be seen as a coarse-time-scale approximation of e-prop. On top of these, our results point to a potential computational benefit of biological non-negative constraint, which could effectively limit the parameter space and promote learning.

## Methods
### Online value-RNN with backprop (oVRNNbp)
We constructed an online value-RNN model based on the previous proposals (*Qian et al., 2025*; *Hennig et al., 2023*) but with several differences. We assumed that the activities of neurons in the RNN at time $t + 1$ were determined by the activities of these neurons and neurons representing observation (cue, reward, or nothing) at time $t$:

$$x(t+1) = f(Ax(t) + Bo(t)),$$ (1.1)

where

$\quad x = (x_j)$: activity of $j$th neuron in the RNN ($j = 1, .., n$)
$\quad o = (o_k)$: activity of $k$th neuron in the observation layer ($k = 1, 2$)
$\quad$ if there was a cue at $t, o(t) = (1\,0)^T$,
$\quad$ if there was a reward at $t, o(t) = (0\,1)^T$,
$\quad$ and otherwise, $o(t) = (0\,0)^T$
$\quad A = (A_{ij})$: recurrent connection strength from $x_j$ to $x_i$
$\quad B = (B_{ik})$: feed-forward connection strength from $o_k$ to $x_i$

$$f(z) = 1/(1 + exp(-z)) - 0.5 :$$ (1.2)

sigmoidal function representing neuronal input–output relation.

The estimated value of the state at $t$ was calculated as

$$v(t) = w^T x(t),$$ (1.3)

where

$$w = (w_j),$$

were the value weights. The error between this estimated value and the true value, $v_{true}(t)$, was defined as

$$\varepsilon(t) = v_{true}(t) - v(t).$$ (1.4)

$$-\partial\left(\varepsilon\left(t\right)^2\right)/\partial_{wj}$$

$$= 2\varepsilon\left(t\right)\partial\varepsilon\left(t\right)/\partial w_j$$

$$= -2\varepsilon\left(t\right)\partial\left(v_{true}\left(t\right) - w^T x\left(t\right)\right)/\partial_{wj}$$

$$= -2\varepsilon\left(t\right)\left(-x_j\left(t\right)\right)$$

$$= 2\varepsilon\left(t\right)x_j\left(t\right)$$

$$\approx 2\delta\left(t\right)x_j\left(t\right)$$

(1.5)

In the last line, since $\varepsilon\left(t\right)$ was unavailable as $v_{true}\left(t\right)$ was unknown, it was approximated by the TD-RPE:

$$\delta\left(t\right) = r\left(t\right) + \gamma v\left(t + 1\right) - v\left(t\right).$$

(1.6)

$-\partial\left(\varepsilon\left(t\right)^2\right)/\partial A_{ij}$ was calculated as follows:

$$-\partial\left(\varepsilon\left(t\right)^2\right)/\partial A_{ij}$$

$$= -2\varepsilon\left(t\right)\partial\left(v_{true}\left(t\right) - w^T x\left(t\right)\right)/\partial A_{ij}$$

$$\approx 2\delta\left(t\right)\partial\left(w^T f\left(A x\left(t - 1\right) + B o\left(t - 1\right)\right)\right)/\partial A_{ij}$$

$$= 2\delta\left(t\right)x_i\left(t - 1\right)\left(0.5 + x_i\left(t\right)\right)\left(0.5 - x_i\left(t\right)\right)w_i$$

(1.7)

Similarly, $-\partial(\varepsilon(t)^2)/\partial B_{ik}$ was calculated as follows:

$$-\partial\left(\varepsilon(t)^2\right)/\partial B_{ik}$$

$$\approx 2\delta(t)o_k(t - 1)\left(0.5 + x_i(t)\right)\left(0.5 - x_i(t)\right)w_i$$

(1.8)

According to these, the online update rule for the value-RNN was determined as follows:

$$w_j \leftarrow w_j + a_{\text{value}}\delta\left(t\right)x_j\left(t\right),$$

(1.9)

$$A_{ij} \leftarrow A_{ij} + a_{\text{RNN}}\delta\left(t\right)x_j\left(t - 1\right)\left(0.5 + x_i\left(t\right)\right)\left(0.5 - x_i\left(t\right)\right)w_i,$$

(1.10)

$$B_{ik} \leftarrow B_{ik} + a_{\text{RNN}}\delta\left(t\right)o_k\left(t - 1\right)\left(0.5 + x_i\left(t\right)\right)\left(0.5 - x_i\left(t\right)\right)w_i,$$

(1.11)

where $a_{\text{value}}$ and $a_{\text{RNN}}$ were the learning rates. In each simulation, the elements of **A** and **B**, as well as the elements of $x$, were initialized to pseudo standard normal random numbers, and the elements of $w$ were initialized to 0.

## Online value-RNN with fixed random feedback (oVRNNrf)

We considered an implementation of the online value-RNN described above in the cortico-basal ganglia-DA system (*Figure 1*):

$x$: activities of neurons in a cortical region with rich recurrent connections
**A**: recurrent connection strengths among $x$
$o$: activities of neurons in a cortical region processing sensory inputs
**B**: feed-forward connection strengths from $o$ to $x$
$f$: sigmoidal relationship from the input to the output of the cortical neurons
$w$: connection strengths from cortical neurons $x$ to a group of striatal neurons
$v$: activity of the group of striatal neurons
$\delta$: activity of a group of DA neurons/released DA

The update rule for $w$ (*Equation 1.9*):

$$w_j \leftarrow w_j + a_{\text{value}}\delta\left(t\right)x_j\left(t\right)$$

could be naturally implemented as cortico-striatal synaptic plasticity, which depends on DA ($\delta(t)$) and pre-synaptic (cortical) neuronal activity ($x_j(t)$). However, an issue emerged in the implementation of the update rules for **A** and **B** (*Equations 1.10 and 1.11*):

$$A_{ij} \leftarrow A_{ij} + a_{\text{RNN}} \delta(t) x_j(t-1) \left(0.5 + x_i(t)\right) \left(0.5 - x_i(t)\right) w_i,$$

$$B_{ik} \leftarrow B_{ik} + a_{\text{RNN}} \delta(t) o_k(t-1) \left(0.5 + x_i(t)\right) \left(0.5 - x_i(t)\right) w_i,$$

Specifically, $w_i$ included in the rightmost of these update rules (for the strengths of cortico-cortical synapses $A_{ij}$ and $B_{ik}$) is the connection strength from cortical neuron $x_i$ to striatal neurons, that is, the strength of the cortico-striatal synapses (located within the striatum), which is considered to be unavailable at the cortico-cortical synapses (located within the cortex).

As mentioned in the Introduction, this is an example of the long-standing difficulty in biological implementation of backprop, and recently a potential solution for this difficulty, that is, replacement of the downstream connection strengths in the update rule for upstream connections with fixed random strengths, has been demonstrated in supervised learning of feed-forward and recurrent networks (*Murray, 2019*; *Lillicrap et al., 2016*; *Wärnberg and Kumar, 2023*). The online value-RNN, which we considered here, differed from supervised learning considered in these previous studies in two ways: (1) it was TD learning, apparent in the approximation of the true error $\varepsilon(t)$ by the TD-RPE $\delta(t)$ in the derivation described above, and (2) it used a scalar error (TD-RPE) rather than a vector error. But we expected that the FA mechanism could still work at least to some extent and explored it in this study. Specifically, we examined a modified online value-RNN with fixed random feedback (oVRNNrf), in which the update rules for **A** and **B** were modified as follows:

$$A_{ij} \leftarrow A_{ij} + a_{\text{RNN}} \delta(t) x_j(t-1) \left(0.5 + x_i(t)\right) \left(0.5 - x_i(t)\right) c_i, \tag{2.1}$$

$$B_{ik} \leftarrow B_{ik} + a_{\text{RNN}} \delta(t) o_k(t-1) \left(0.5 + x_i(t)\right) \left(0.5 - x_i(t)\right) c_i, \tag{2.2}$$

where $w_i$ in the update rules of the online value-RNN with backprop (oVRNNbp) was replaced with a fixed random parameter $c_i$. Notably, these modified update rules for the cortico-cortical connections **A** and **B** required only pre-synaptic activities ($x_j(t-1)$, $o_k(t-1)$), post-synaptic activities ($x_i(t)$), TD-RPE-representing DA ($\delta(t)$), and fixed random strengths ($c_i$), which would all be available at the cortico-cortical synapses given that VTA DA neurons project not only to the striatum but also to the cortex and random $c_i$ could be provided by intrinsic heterogeneity. In each simulation, the elements of $c$ were initialized to pseudo standard normal random numbers.

## Revised online value-RNN models with further biological constraints

In the later part of this study, we examined revised online value-RNN models with further biological constraints. Specifically, we considered models in which the value weights and the activities of neurons in the RNN were constrained to be non-negative. In order to do so, the update rule for $w$ was modified to:

$$w_j \leftarrow \max \left(0, w_j + a_{\text{value}} \delta(t) x_j(t)\right), \tag{3.1}$$

where $\max(q_1, q_2)$ returned the maximum of $q_1$ and $q_2$. Also, the sigmoidal input–output function was replaced with

$$f(z) = 1/(1 + \exp(-z)), \tag{3.2}$$

and the elements of $x$ were initialized to pseudo uniform [0 1] random numbers. The backprop-based update rules for **A** and **B** in oVRNNbp were replaced with

$$A_{ij} \leftarrow A_{ij} + a_{\text{RNN}} \delta(t) x_j(t-1) x_i(t) \left(1 - x_i(t)\right) w_i, \tag{3.3}$$

$$B_{ik} \leftarrow B_{ik} + a_{\text{RNN}} \delta(t) o_k(t-1) x_i(t) \left(1 - x_i(t)\right) w_i. \tag{3.4}$$

We referred to the model with these modifications to oVRNNbp as oVRNNbp-rev.

As a revised online value-RNN with fixed random feedback (oVRNNrf), in addition to the above-mentioned modifications of the update of $w$, the sigmoidal input-output function, and the initialization of $x$, the fixed random feedback $c$ was assumed to be non-negative. Specifically, the elements of $c$

were set to pseudo uniform [0 1] random numbers. Moreover, the update rules for **A** and **B** were replaced with

(when $x_i(t) \leq 0.5$)

$$A_{ij} \leftarrow A_{ij} + a_{\text{RNN}}\,\delta(t)\,x_j(t-1)\,x_i(t)\left(1 - x_i(t)\right)c_i, \tag{3.5}$$

$$B_{ik} \leftarrow B_{ik} + a_{\text{RNN}}\,\delta(t)\,o_k(t-1)\,x_i(t)\left(1 - x_i(t)\right)c_i. \tag{3.6}$$

(when $x_i(t) > 0.5$)

$$A_{ij} \leftarrow A_{ij} + 0.25\,a_{\text{RNN}}\,\delta(t)\,x_j(t-1)\,c_i, \tag{3.7}$$

$$B_{ik} \leftarrow B_{ik} + 0.25\,a_{\text{RNN}}\,\delta(t)\,o_k(t-1)\,c_i, \tag{3.8}$$

so that the originally non-monotonic dependence on $x_i(t)$ (post-synaptic activity) became monotonic + saturation (**Figure 5B**). These update rules with non-negative $c_i$ could be said to be Hebbian with additional modulation by TD-RPE (Hebbian under positive TD-RPE) (see Appendix 1.2 for possible consideration of behavioral time-scale synaptic plasticity (BTSP) in our models). We referred to the model with these modifications to oVRNNrf as oVRNNrf-bio. In the right panels of **Figure 6E, F**, we also examined the model where the modified update rules of oVRNNrf-bio were changed back to the original ones, referred to as oVRNNrf-rev. In some simulations in **Figure 8E–G**, we examined a modified oVRNNrf-bio with a slight decay (forgetting) of value weights, in which each element of $w$ decayed at every time step:

$$w_i \leftarrow (1 - dr)\,w_i, \tag{3.9}$$

where $dr$ was the decay rate per time step and was set to 0.001 or 0.002.

We further examined extensions of oVRNNbp-rev and oVRNNrf-bio, referred to as oVRNNbp-rev-ei and oVRNNrf-bio-ei, which incorporated excitatory E-units and inhibitory I-units (**Figure 9A**). Based on biological suggestions (see the Results), we made the following assumptions. Each E-unit received inputs from the observation units $o$ (connections: $\mathbf{B_E}$), all the E-units (connections: $\mathbf{A_E}$), and a particular I-unit (with a strength $h$), and projected to the striatal value unit (connections: $w$). Each I-unit received inputs from the observation units $o$ (connections: $\mathbf{B_I}$) and all the E-units (connections: $\mathbf{A_I}$). Excitation from the observation units and E-units to E- and I-units took one time step, whereas I → E inhibition operated within a time step. The activation function for E-unit and plasticity rules for connections from/to E-units were the same as those for the RNN unit in the original models. I-unit had a linear activation function, and there was no plasticity for connections from/to I-units. The update rule for $w$ was the same as the original one with the activity of the RNN units replaced with the activity of E-units. Equations for the activities of E-units and I-units, $x_E$ and $x_I$, are given as follows:

$$\boldsymbol{x}_{\mathbf{I}}(t+1) = \boldsymbol{A_I}\boldsymbol{x}_{\mathbf{E}}(t) + \boldsymbol{B_I}\boldsymbol{o}(t), \tag{3.10}$$

$$\boldsymbol{x}_{\mathbf{E}}(t+1) = \boldsymbol{f}\left(\boldsymbol{A_E}\boldsymbol{x}(t) + \boldsymbol{B_E}\boldsymbol{o}(t) - h\boldsymbol{x}_{\mathbf{I}}(t+1)\right), \tag{3.11}$$

where $h$ was set to 1. The elements of $\mathbf{A_I}$, $\mathbf{B_I}$, $\mathbf{A_E}$, and $\mathbf{B_E}$ were initialized to be non-negative: $\max(0, 3+z)$, where $z$ was a pseudo standard normal random number. The elements of $x_E$ were initialized to pseudo uniform [0 1] random numbers, and the initial values of $x_I$ were determined according to the abovementioned equation.

## Incorporation of action selection

We considered extensions of oVRNNbp-rev and oVRNNrf-bio that incorporated an actor-critic architecture, referred to as oVRNNbp-rev-ac and oVRNNrf-bio-ac (**Figure 11A**). Each RNN unit additionally connected to additional two units representing the action-values of action 1 and action 2 ($q_1$ and $q_2$):

$$\boldsymbol{q}(t) = \mathbf{U}\boldsymbol{x}(t), \tag{4.1}$$

where $U = (u_{kj})$ consisted of two row vectors that represented the preferences of the two actions. At the time step next to cue presentation, one action was selected in a soft-max manner based on the action values. Specifically, action $k$ was selected with the probability of

$$\exp(\beta q_k)/(\exp(\beta q_1) + \exp(\beta q_2)), \tag{4.2}$$

where $\beta$ was the inverse temperature parameter, set to 1 or 2, representing the degree of exploitation over exploration. The selected action was then informed to the RNN units. Specifically, the observation layer had two additional elements ($o'$ in **Figure 11A**) corresponding to the two actions. These elements became 1 when the corresponding action was selected and 0 otherwise. The preference of selected action $k$ was updated by using the TD-RPE $\delta(t)$ as follows:

$$u_{kj} \leftarrow u_{kj} + a_{\mathrm{pref}}\delta(t)\, x_j(t), \tag{4.3}$$

where $a_{\mathrm{pref}}$ was the learning rate. In order to prevent unbounded increase of action preference, we assumed a slight decay of all the action preferences at every time step:

$$u_{kj} \leftarrow (1 - dr)\, u_{kj}, \tag{4.4}$$

where $dr$ was the decay rate per time step and was set to 0.001. $u_{kj}$ (i.e., the elements of **U**) were initialized to 0. The connection weights from the action-observation units $o'$, as well as from $o$ and the RNN units, onto the RNN units were initialized to pseudo standard normal random numbers.

## Simulation of the tasks

In the Pavlovian cue–reward association task, at time 1 of each trial, cue observation was received by the RNN, and at time 4, reward observation was received. The trial was pseudo-randomly ended at time 7, 8, 9, or 10, and the next trial started from the next time step (i.e., ITI was 4, 5, 6, or 7 time steps with equal probabilities). Reward size was $r=1$. We also conducted simulations with longer cue–reward delays, in which reward was given at time 5, 6, or 7, and the end of trial was shifted accordingly. The tasks with probabilistic structures (tasks 1 and 2) were implemented in the same way except that reward timing was not time 4 but time 3 or 5 with equal probabilities, specifically, 50% and 50% in task 1 and 30% and 30% in task 2, and there was no reward in the remaining 40% of trials in task 2. The tasks with action selection were also implemented in the same way except that size 2 reward was received, that is, the reward term in the TD-RPE calculation as well as the reward-corresponding element of the observation inputs were set to 2, at time 4 when action 1 was selected, whereas size 1 reward was received at time 4 (in the first choice task) or time 3 (in the second choice task) when action 2 was selected.

The cue or reward state/timing, mentioned in the text and marked in the figures, was defined to be the timing when the RNN received the cue or reward observation, respectively. Specifically, if $o(t) = (1\,0)^T$ or $o(t) = (0\,1)^T$ at time $t$, $t + 1$ was defined to be a cue or reward timing, respectively. For the agents with punctate state representation, which is also referred to as the complete serial compound representation (**Montague et al., 1996**; **Sutton and Barto, 2018**; **Ludvig et al., 2012**), each timing from a cue in the tasks was represented by a 10-dimensional one-hot vector, starting from $(1\,0\,0\ldots 0)^T$ for the cue state, with the next state $(0\,1\,0\ldots 0)^T$ and so on.

In the simulations of the cue–reward association task with distractor cue, the observation units $o = (o_k)$ had an additional element $o_3$ (**Figure 10A**), which was 1 at the time steps where distractor cue was present and 0 otherwise. We examined four cases, in which the probability of the presence of distractor cue at every time step throughout the task was 0, 0.1, 0.2, and 0.3 (**Figure 10B–E**).

Learning rates were set as follows. For the models with punctate state representation, $a_{\mathrm{value}} = 0.1$. For oVRNNbp, oVRNNrf, and the model with untrained RNN compared with these two, $a_{\mathrm{RNN}} = 0.1$ and $a_{\mathrm{value}} = 0.1/(n/7)$. For oVRNNbp-rev, oVRNNrf-bio, and the models with untrained RNN compared with these two, $a_{\mathrm{RNN}} = 0.1$ and $a_{\mathrm{value}} = a_{\mathrm{pref}} = 0.1/(n/12)$ except for the right panels of **Figure 6J**, and $a_{\mathrm{RNN}} = 0.05$ and $a_{\mathrm{value}} = 0.05/(n/12)$ for the right panels of **Figure 6J**. Time discount factor ($\gamma$) was set to 0.8.

## Estimation of true state/timing values

As for the Pavlovian cue–reward association task, we defined states after agent's receival of cue information by relative timings from the cue and estimated their (true) values by simulations according to the definition of state value. We generated a sequence of cues and rewards corresponding to 1000 trials with the ITI after the first trial, $\mathrm{ITI}_1$, fixed to one of the possible lengths (4, 5, 6, or 7 time steps), and calculated cumulative discounted future rewards within the sequence:

$$\Sigma_{t\_\mathrm{rew}}\,(r\gamma^{t-\mathrm{rew}}),$$

where $t_{rew}$ denotes the time step of each reward counted from the starting state, starting from +1,..., and +3 + $ITI_1$ time steps from a cue (the last one corresponded to the cue timing of the next trial). For each case where $ITI_1$ = 4, 5, 6, or 7, we repeated this 1000 times, generating 1000 sequences (i.e., 1000 simulations of 1000 trials), with different sets of pseudo-random numbers, and calculated the average over these 1000 sequences (we refer to these as $ITI_1$-specific values). We estimated the value of each state of +1,..., and +7 time steps from cue (i.e., −2,...,+4 time steps from reward) by taking the average of the $ITI_1$-specific values for four possible $ITI_1$.

We also estimated the true values of the cue timing and one and two timing(s) before it in the following way; these values could not be estimated in the abovementioned way because the agent should not know the length of ITI (i.e., when ITI ends) until receiving cue information at the cue timing. In the case where ITI is in fact 4 time steps, until receiving the next cue, the agent should think that ITI can be 4, 5, 6, or 7 time steps with equal probabilities (1/4 for each). Thus, the value of next cue timing and one and two timing(s) before it should be the average of the four $ITI_1$-specific values of +4, +3, and +2 time steps from reward. Similarly, in the case where ITI is in fact 5 time steps, until the previous time step of the next cue, the agent should think that ITI can be 4, 5, 6, or 7 time steps with equal probabilities (1/4 for each). Thus, the value of one and two timing(s) before next cue should be the average of the four $ITI_1$-specific values of +4 and +3 time steps from reward. On the other hand, at the timing of the next cue, the agent should think that ITI can be 5, 6, or 7 (but not 4) time steps with equal probabilities (1/3 for each). Thus, the value of next cue timing should be the average of the three $ITI_1$-specific values (for $ITI_1$ = 5, 6, or 7) of +5 time steps from reward. Similar considerations can be made for the cases where ITI is in fact 6 or 7 time steps. And then, the 'true' value of (next) cue timing can be calculated as the average of the values of next cue timing in the cases where ITI is in fact 4, 5, 6, or 7 time steps. Using these estimated true state values, we calculated TD-RPE at each state/timing (−2, −1, ..., and +5 time steps from cue). True state/timing values in the cases where the cue–reward delay was 4, 5, or 6 time steps were estimated in the same way.

We also estimated true state/timing values for tasks 1 and 2 that had probabilistic structures. As for task 1, we first estimated the values of each timing in each of the trial types (*Figure 4Ba*, left), in which reward was given at early (2 time steps after cue) or late (4 time steps after cue) timing, in the same manner (but using 10000 rather than 1000 simulations for each condition) as done for the cue–reward association task mentioned above (values of the cue timing and the one and two timing(s) before cue after each trial type were also estimated). Then, based on the agent's belief about trial types (*Figure 4Bb*, left), we defined the following states: +1 and +2 time steps from cue (i.e., states visited [entered] before knowing whether reward was given at the early timing [= +2 time step from cue]), +3, 4, 5, and 6 time steps from cue after reception of reward at the early timing, and +3, 4, 5, and 6 time steps from cue after no reception of reward at the early timing (*Figure 4Bc*, left-top). We calculated the true values of these states and also of the cue timing and one and two timing(s) before cue (*Figure 4Bc*, left-bottom) by taking (mathematical) expected value of the abovementioned estimated value of each timing in each trial type. Using these true values, we calculated TD-RPE (*Figure 4C*, left).

As for task 2, we first estimated the values of each timing in each of the trial types (*Figure 4Ba*, right), in which reward was given at early (2 time steps after cue) or late (4 time steps after cue) timing or was not given. Then, based on the agent's belief about trial types (*Figure 4Bb*, right), we defined the following states: +1 and +2 time steps from cue (i.e., states visited [entered] before knowing whether reward was given at the early timing), +3, 4, 5, and 6 time steps from cue after reception of reward at the early timing, +3 and 4 time steps from cue after no reception of reward at the early timing (states visited [entered] before knowing whether reward was given at the late timing [= +4 time step from cue]), +5 and 6 time steps from cue after reception of reward at the late timing, and +5 and 6 time steps from cue after no reception of reward at both early and late timings (*Figure 4Bc*, right-top). We estimated the true values of these states and also of the cue timing and one and two timing(s) before cue (*Figure 4Bc*, right-bottom) in the same manner as for task 1, and using these true values, we calculated TD-RPE (*Figure 4C*, right).

## Analyses, software, and code availability

SEM was approximated by SD/√$N$ (number of samples). Cohen's $d$ using an average variance was calculated as (difference in the means)/(square root of the average of variances). Linear regression,

PCA, Wilcoxon rank sum test, and $t$-tests were conducted by using R (functions lm, prcomp, wilcox. exact (in package exactRankTests), and t.test). The difference in the Wilcoxon rank sum test and $t$-tests was reported when $p < 0.05$. Simulations were conducted by using MATLAB, and pseudo-random numbers were implemented by using rand, randn, and randperm functions. The codes for simulations and analyses are available at GitHub (https://github.com/kenjimoritagithub/oVRNN1, copy archived at *Morita, 2025*).

## Acknowledgements

The authors thank Dr. Kenji Doya for valuable suggestions. KM was supported by Grants-in-Aid for Scientific Research 23H03295, 23K27985, and 25H02594 from Japan Society for the Promotion of Science (JSPS) and the Naito Foundation. AyK was supported by JSPS Overseas Research Fellowships. ArK was partially funded by Digital Futures (KTH) grant and StratNeuro SRA.

## Additional information

### Funding

| Funder | Grant reference number | Author |
|---|---|---|
| Japan Society for the Promotion of Science | 23H03295 | Kenji Morita |
| Japan Society for the Promotion of Science | 23K27985 | Kenji Morita |
| Japan Society for the Promotion of Science | 25H02594 | Kenji Morita |
| Naito Foundation | | Kenji Morita |
| Japan Society for the Promotion of Science | Overseas Research Fellowships | Ayaka Kato |
| Digital Futures | | Arvind Kumar |
| StratNeuro SRA | | Arvind Kumar |

The funders had no role in study design, data collection, and interpretation, or the decision to submit the work for publication.

### Author contributions

Takayuki Tsurumi, Formal analysis, Investigation, Writing – review and editing; Ayaka Kato, Investigation, Writing – review and editing; Arvind Kumar, Writing – review and editing; Kenji Morita, Conceptualization, Formal analysis, Investigation, Writing – original draft, Writing – review and editing

### Author ORCIDs

Ayaka Kato https://orcid.org/0000-0002-6306-6600
Arvind Kumar https://orcid.org/0000-0002-8044-9195
Kenji Morita https://orcid.org/0000-0003-2192-4248

Reviewer #1 (Public review): https://doi.org/10.7554/eLife.104101.4.sa1
Reviewer #2 (Public review): https://doi.org/10.7554/eLife.104101.4.sa2
Reviewer #3 (Public review): https://doi.org/10.7554/eLife.104101.4.sa3
Author response https://doi.org/10.7554/eLife.104101.4.sa4

## Additional files

### Supplementary files

MDAR checklist

## Data availability

The codes for simulations and analyses are available at GitHub (https://github.com/kenjimoritagithub/oVRNN1, copy archived at *Morita, 2025*).

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

# Appendix 1

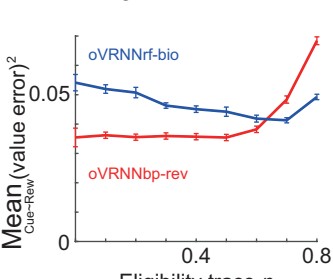
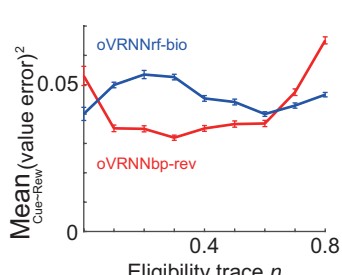

Learning rate: default Learning rate: halved

**Appendix 1—figure 1.** Performance (mean squared value-error from cue to reward) of modified oVRNNbp-rev (red lines) and oVRNNrf-bio (blue lines) incorporating eligibility trace in the Pavlovian cue–reward association task with 6 time steps cue–reward delay, with the default learning rates (left panel) or halved learning rates (right panel) used in *Figure 6J*, left and right, respectively. The eligibility-trace parameter $\eta$ was varied as indicated by the horizontal axis. The data at $\eta$ = 0 are the results for the original models without eligibility trace (the same data as those shown in *Figure 6*), for which 100 simulations were executed. For the other cases, we executed 1000 simulations in order to precisely examine changes in the performance.

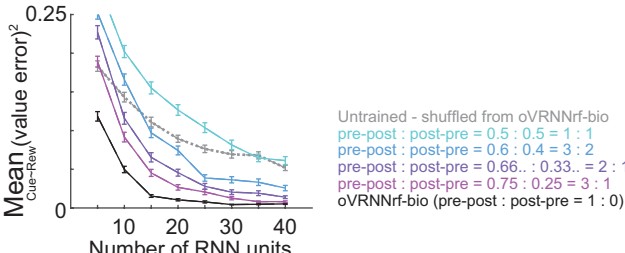

**Appendix 1—figure 2.** Performance (mean squared value-error from cue to reward) of modified oVRNNrf-bio incorporating behavioral time-scale synaptic plasticity (BTSP) in the Pavlovian cue–reward association task with 3 time steps cue–reward delay. In the modified model, the update rule for the weights on the RNN units was modified so that the update could depend on both pre → post and post → pre activity pairings. The proportion of update induced by pre → post activity pairing (parameter $k_{\text{pre}\rightarrow\text{post}}$) was varied to 1 (corresponding to the original oVRNNrf-bio), 0.75, 2/3(=0.66..), 0.6, or 0.5, indicated by solid lines with different colors. The gray dotted line indicates the performance of an agent having an untrained RNN with connections shuffled from learned original oVRNNrf-bio. Other configurations are the same as those for *Figure 6E*, left. Other parameters were set as follows. Learning rates: aRNN = 0.1 and avalue = 0.1/(n/12). Time discount factor: $\gamma$ = 0.8. Number of trials: 1500. Number of simulations: 100 for each condition.

## 1.1 Eligibility trace

As mentioned in the Introduction and Discussion, our model has relations to other models, including the algorithms named RFLO (random feedback local online) (*Murray, 2019*) and e-prop (*Bellec et al., 2020*). In the derivation of RFLO (*Murray, 2019*), initially, the partial derivative of the across-time sum of squared errors with respect to each RNN weight was taken (as in the exact RTRL (Real-Time Recurrent Learning)), and then in order to make the update rule local, the parts depending on the other weights were omitted, while the local 'eligibility traces' remained. e-prop (*Bellec et al., 2020*) also resulted in the terms interpreted as eligibility traces. By contrast, we started from the partial derivative of the squared error at the current time rather than its across-time sum, and the resulting update rule does not contain eligibility trace. Our model could still solve credit assignment at least to some extent because we use TD error and TD learning itself (even TD(0) without eligibility trace) is a solution for credit assignment as we mentioned in the Discussion.

Based on these considerations, we conjectured that adding eligibility trace to our update rule could possibly improve the learnability of our model, especially in the case with long cue–reward delay, and we tested it. We considered modified models, in which the terms depending on the

pre-synaptic and post-synaptic activities in the update rules for the connections on the RNN units were replaced with the terms including eligibility traces that were locally updated and exponentially decayed. Specifically, we examined modified oVRNNbp-rev and oVRNNrf-bio, in which the following part in the update rules for elements of **A** and **B**:

$X_{ij} = x_j (t - 1) x_i (t) (1 - x_i (t))$ in oVRNNbp-rev, or $X_{ij} = x_j (t - 1) x_i (t) (1 - x_i (t)) (x_i (t) \leq 0.5)$ or $0.25 x_j (t - 1) (x_i (t) > 0.5)$ in oVRNNrf-bio was replaced with $(1 - \eta) Z_{ij}$, where $Z_{ij}$ was updated at every time step as follows:

$Z_{ij} \leftarrow \eta Z_{ij} + x_j (t - 1) x_i (t) (1 - x_i (t))$ in oVRNNbp-rev, or $Z_{ij} \leftarrow \eta Z_{ij} + x_j (t - 1) x_i (t) (1 - x_i (t)) (x_i (t) \leq 0.5)$ or $0.25 x_j (t - 1) (x_i (t) > 0.5)$ in oVRNNrf-bio was a parameter indicating the length (time scale) of eligibility trace.

We simulated the Pavlovian cue–reward association task with 6 time steps cue–reward delay, varying the parameter $\eta$. As a result, in the model with random feedback, oVRNNrf-bio, with 40 RNN units and the default learning rates that we used, eligibility trace indeed improved the performance, though not drastically (the blue line in the left panel of *Appendix 1—figure 1*), while in the model with symmetric feedback (oVRNNbp-rev), there was no improvement (the red line in the left panel). In the cases with the learning rates halved (used in *Figure 6J*, right), the results look more complicated (the right panel).

## 1.2 Behavioral time-scale synaptic plasticity

We originally assumed that update of the connections onto the RNN units is implemented by Hebbian synaptic plasticity. However, considering this point further, we have realized that Hebbian plasticity, or at least a typical form of it, namely, the spike-timing-dependent-plasticity (STDP) (*Markram et al., 1997*; *Bi and Poo, 1998*) may not be able to implement the update of the RNN connections in our model. This is because while the single time step in our model was assumed to correspond to several hundreds of milliseconds, the time scale of STDP is much shorter, up to several tens of milliseconds. Besides, measurable degrees of STDP would be caused only after repetitive pairing of pre- and post-synaptic spikes, whereas our non-spiking rate-based model has no description of such repetitive pairing. There have been proposals on how such a gap between the time scale of STDP and behavioral time scale could be bridged (*George et al., 2023*; *Bono et al., 2023*; *Drew and Abbott, 2006*), and with such mechanisms, STDP could still potentially implement the update of the RNN connections in our model. However, it may be more likely to be implemented by a different form of synaptic plasticity named the behavioral time-scale synaptic plasticity (BTSP) (*Bittner et al., 2017*; *Caya-Bissonnette et al., 2023*).

BTSP, originally found in hippocampal CA1 synapses (*Bittner et al., 2017*), is induced by pairing of pre- and post-synaptic activity, similar to STDP. However, different from STDP, the time scale of BTSP is much longer, up to seconds, and also BTSP can be induced with only a few, or even a single, pairing. These features could better fit the update of the RNN connections in our model. Recently, *Caya-Bissonnette et al., 2023*, BTSP was found also in the neocortex, in particular, the prefrontal cortex, which receives rich DA projections and is thus a primarily hypothesized brain region for the RNN in our model. Compared to hippocampal BTSP, prefrontal BTSP has somewhat shorter time scale, several hundreds of millisecond difference between pre- and post-synaptic activity, for potentiation, and also differs in that pairing within a time window of 100–300 ms does not cause potentiation (instead causing slight depression) (*Caya-Bissonnette et al., 2023*).

In terms of time scale, BTSP appears to be good for implementing the update of the RNN connections in our model. However, two points need consideration. First, bidirectional modulation of BTSP by positive/negative DA signals will be required, but it has not yet been experimentally shown. Second, in both hippocampal (*Bittner et al., 2017*) and prefrontal (*Caya-Bissonnette et al., 2023*) BTSP, potentiation is caused not only by the pre → post order but also by the post → pre order of activity pairing, different from STDP. As for hippocampal BTSP, there is still an asymmetry, that is, potentiation occurs more/longer for pre → post pairing than for post → pre pairing (Figure 3D of *Bittner et al., 2017*). Similar asymmetry was not reported for prefrontal BTSP (*Caya-Bissonnette et al., 2023*), although asymmetry could still possibly appear due to additional factors in vivo. The

update rule for the RNN connections in our model, inherited from the original derivation according to the gradient descent, assumes potentiation for pre → post pairing but not for post → pre pairing under positive TD-RPE. It thus remains elusive whether learning could still occur in our model when not only pre → post pairing but also post → pre pairing induces potentiation under positive TD-RPE.

We examined this by simulations of the Pavlovian cue–reward association task with 3 time steps cue–reward delay with modified oVRNNrf-bio, in which update of the connections on the RNN units was induced by pre → post and post → pre activity pairings with $k_{pre \rightarrow post} : 1 - k_{pre \rightarrow post}$ ratio, with $k_{pre \rightarrow post}$ varied from 1 (corresponding to the original oVRNNrf-bio), 0.75 (3:1 ratio), 0.66 (2:1 ratio), 0.6 (3:2 ratio), to 0.5 (1:1 ratio, i.e., symmetric). More specifically, we examined modified oVRNNrf-bio, in which the update rules for **A** and **B** (*Equations 3.5–3.8*) were replaced with the followings:

$$\Delta A_{ij_{\text{pre} \rightarrow \text{post}}} = a_{\text{RNN}} \delta(t) x_j(t-1) x_i(t) (1 - x_i(t)) c_i (x_i(t) \leq 0.5)$$

$$\text{or } 0.25 a_{\text{RNN}} \delta(t) x_j(t-1) c_i (x_i(t) > 0.5)$$

$$\Delta B_{ik_{\text{pre} \rightarrow \text{post}}} = a_{\text{RNN}} \delta(t) o_k(t-1) x_i(t) (1 - x_i(t)) c_i (x_i(t) \leq 0.5)$$

$$\text{or } 0.25 a_{\text{RNN}} \delta(t) o_k(t-1) c_i (x_i(t) > 0.5)$$

$$\Delta A_{ij\_\text{post} \rightarrow \text{pre}} = a_{\text{RNN}} \delta(t) x_j(t) x_i(t-1) (1 - x_i(t-1)) c_i (x_i(t-1) \leq 0.5)$$

$$\text{or } 0.25 a_{\text{RNN}} \delta(t) x_j(t) c_i (x_i(t-1) > 0.5)$$

$$\Delta B_{ik\_\text{post} \rightarrow \text{pre}} = a_{\text{RNN}} \delta(t) o_k(t) x_i(t-1) (1 - x_i(t-1)) c_i (x_i(t-1) \leq 0.5)$$

$$\text{or } 0.25 a_{\text{RNN}} \delta(t) o_k(t) c_i (x_i(t-1) > 0.5)$$

$$A_{ij} \leftarrow A_{ij} + k_{\text{pre} \rightarrow \text{post}} \Delta A_{ij_{\text{pre} \rightarrow \text{post}}} + (1 - k_{\text{pre} \rightarrow \text{post}}) \Delta A_{ij\_\text{post} \rightarrow \text{pre}}$$

$$B_{ij} \leftarrow B_{ij} + k_{\text{pre} \rightarrow \text{post}} \Delta B_{ij_{\text{pre} \rightarrow \text{post}}} + (1 - k_{\text{pre} \rightarrow \text{post}}) \Delta B_{ij\_\text{post} \rightarrow \text{pre}}$$

where $k_{\text{pre} \rightarrow \text{post}}$ is a parameter indicating the proportion of update induced by pre → post activity pairing and was set to 1 (corresponding to the original oVRNNrf-bio), 0.75, 2/3(=0.66..), 0.6, or 0.5.

The results, mean squared errors in the state values from cue to reward, are shown in Figure A2. As shown in the figure, the model performed better than the untrained control with connections shuffled from learned original oVRNNrf-bio even when the update of the RNN connections was induced by not only pre → post but also post → pre activity pairings if there was at least a modest level of asymmetry (3:2) and the number of RNN units was not small (25 or above).

#Addition of synaptic update by post → pre pairing up to a certain extent thus does not totally ruin the performance of our model. But it still causes a degradation, rather than an improvement, in terms of this performance measure, as would be expected. However, the addition of synaptic update by post → pre pairing could potentially provide the model with other features. Specifically, given that potentiation by pre → post pairing could cause self-organization of the successor representation (SR) (*George et al., 2023*; *Bono et al., 2023*), potentiation by post → pre pairing could potentially cause self-organization of the opposite, predecessor representation (PR) (cf., *Bailey and Mattar, 2022*). Then, having both pre → post potentiation and post → pre potentiation could achieve a formation of state representation that has features of both SR and PR. Intriguingly, recent studies (*Jeong et al., 2022*; *Garr et al., 2024*; *Floeder et al., 2024*) suggest that mesolimbic DA signals 'adjusted net contingency for causal relations (ANCCR)', which #combines SR and PR. Whether BTSP with both pre → post potentiation and post → pre potentiation contributes to its mechanism may be interesting to address in future studies.

