## [Editor Report · eLife Assessment]

In this **important** study, the authors model reinforcement-learning experiments using a recurrent neural network. The work examines if the detailed credit assignment necessary for back-propagation through time can be replaced with random feedback. The authors provide **solid** evidence that the solution is adequate within relatively simple tasks.

---

## [Referee Report · Reviewer #1 (Public review)]

Summary:

Can a plastic RNN serve as a basis function for learning to estimate value. In previous work this was shown to be the case, with a similar architecture to that proposed here. The learning rule in previous work was back-prop with an objective function that was the TD error function (delta) squared. Such a learning rule is non-local as the changes in weights within the RNN, and from inputs to the RNN depends on the weights from the RNN to the output, which estimates value. This is non-local, and in addition, these weights themselves change over learning. The main idea in this paper is to examine if replacing the values of these non-local changing weights, used for credit assignment, with random fixed weights can still produce similar results to those obtained with complete bp. This random feedback approach is motivated by a similar approach used for deep feed-forward neural networks.

This work shows that this random feedback in credit assignment performs well but is not as well as the precise gradient-based approach. When more constraints due to biological plausibility are imposed performance degrades. These results are consistent with previous results on random feedback.

Strengths:

The authors show that random feedback can approximate well a model trained with detailed credit assignment.

The authors simulate several experiments including some with probabilistic reward schedules and show results similar to those obtained with detailed credit assignments as well as in experiments.

The paper examines the impact of more biologically realistic learning rules and the results are still quite similar to the detailed back-prop model.

---

## [Referee Report · Reviewer #2 (Public review)]

Summary:

Tsurumi et al. show that recurrent neural networks can learn state and value representations in simple reinforcement learning tasks when trained with random feedback weights. The traditional method of learning for recurrent network in such tasks (backpropogation through time) requires feedback weights which are a transposed copy of the feed-forward weights, a biologically implausible assumption. This manuscript builds on previous work regarding "random feedback alignment" and "value-RNNs", and extends them to a reinforcement learning context. The authors also demonstrate that certain non-negative constraints can enforce a "loose alignment" of feedback weights. The author's results suggest that random feedback may be a powerful tool of learning in biological networks, even in reinforcement learning tasks.

Strengths:

The authors describe well the issues regarding biologically plausible learning in recurrent networks and in reinforcement learning tasks. They take care to propose networks which might be implemented in biological systems and compare their proposed learning rules to those already existing in literature. Further, they use small networks on relatively simple tasks, which allows for easier intuition into the learning dynamics.

Weaknesses:

The principles discovered by the authors in these smaller networks are not applied to larger networks or more complicated tasks with long temporal delays (>100 timesteps), so it remains unclear to what degree these methods can scale or can be used more generally.

---

## [Referee Report · Reviewer #3 (Public review)]

Summary:

The paper studies learning rules in a simple sigmoidal recurrent neural network setting. The recurrent network has a single layer of 10 to 40 units. It is first confirmed that feedback alignment (FA) can learn a value function in this setting. Then so-called bio-plausible constraints are added: (1) when value weights (readout) is non-negative, (2) when the activity is non-negative (normal sigmoid rather than downscaled between -0.5 and 0.5), (3) when the feedback weights are non-negative, (4) when the learning rule is revised to be monotic: the weights are not downregulated. In the simple task considered all four biological features do not appear to impair totally the learning.

Strengths:

(1) The learning rules are implemented in a low-level fashion of the form: (pre-synaptic-activity) x (post-synaptic-activity) x feedback x RPE. Which is therefore interpretable in terms of measurable quantities in the wet-lab.

(2) I find that non-negative FA (FA with non negative c and w) is the most valuable theoretical insight of this paper: I understand why the alignment between w and c is automatically better at initialization.

(3) The task choice is relevant, since it connects with experimental settings of reward conditioning with possible plasticity measurements.

---

## [Author Response]

The following is the authors’ response to the previous reviews.

**Reviewer #1 (Public Review):**

We thank the reviewer for the positive feedback on the work. The reviewer has raised two weaknesses and in the following we discuss how those can be addressed.

Weaknesses:The impact of the article is limited by using a network with discrete time- steps, and only a small number of time steps from stimulus to reward. They assume that each time step is on the order of hundreds of ms. They justify this by pointing to some slow intrinsic mechanisms, but they do not implement these slow mechanisms is a network with short time steps, instead they assume without demonstration that these could work as suggested. This is a reasonable first approximation, but its validity should be explicitly tested.

Our goal here was to give a proof of concept that online random feedback is sufficient to train an RNN to estimate value. Indeed, it is important to show that the idea works in a model where the slow mechanisms are explicitly implemented. However, this is a non-trivial task and desired to be addressed in future works.

As the delay between cue and reward increases the performance decreases. This is not surprising given the proposed mechanism, but is still a limitation, especially given that we do not really know what a is the reasonable value of a single time step.

In reply to this comment and the other reviewer's related comment, we have conducted two sets of additional simulations, one for examining incorporation of eligibility traces, and the other for considering (though not mechanistically implementing) behavioral time-scale synaptic plasticity (BTSP). We have added their results to the revised manuscript as Appendix. We think that the results addressed this point to some extent while how longer cue-reward delay can be learnt by elaboration of the model remains as a future issue.

**Reviewer #2 (Public Review):**

We thank the reviewer for the positive feedback on the work. The reviewer gave comments on our revisions, and here we discuss how those can be addressed.

Comments on revisions: I would still want to see how well the network learns tasks with longer time delays (on the order of 100 or even 1000 timesteps). Previous work has shown that random feedback struggles to encode longer timescales (see Murray 2019, Figure 2), so I would be interested to see how that translates to the RL context in your model.

We would like to note that in Murray et al 2019 the random feedback per se appeared not to be primarily responsible for the difficulty in encoding longer timesclaes. In the Figure 2d (Murray 2019), the author compared his RFLO (random feedback local online) and BPTT with two intermediate algorithms, which incorporated either one of the two approximations made in RFLO: (i) random feedback instead of symmetric feedback, and (ii) omittance of non-local effect (i.e., dependence of the derivative of the loss with respect to a given weight on the other weights). The performance difference between RFLO and BPTT was actually mostly explained by (ii), as the author mentioned "The results show that the local approximation is essentially fully responsible for the performance difference between RFLO and BPTT, while there is no significant loss in performance due to the random feedback alone. (Line 6-8, page 7 of Murray, 2019, eLife)".

Meanwhile, regarding the difference in the performance of the model with random feedback vs the model with symmetric feedback in our settings, actually it appeared (already) in the case with 6 time-steps or less (the biologically constrained model with random feedback performed worse: Fig. 6J, left).

In practice, our model, either with random or symmetric feedback, would not be able to learn the cases with very long delays. This is indeed a limitation of our model. However, our model is critically different from the model of Murray 2019 in that we use RL rather than supervised learning and we use a scalar bootstrapped (TD) reward-prediction-error rather than the true output error. We would think that these differences may be major reasons for the limited learning ability of our model.

Regarding the feasibility of the model when tasks involve longer time delays: Indeed this is a problem and the other reviewers have also raised the same point. Our model can be extended by incorporating either a kind of eligibility trace (similar one to those contained in RFLO and e-prop) or behavioral time-scale synaptic plasticity (BTSP), and we have added the results of simulations incorporating each to the revised manuscript as Appendix. But how longer cue-reward delay can be learnt by elaboration of the model remains as a future issue.

**Reviewer #3 (Public Review):**
Comments on revisions: Thank you for addressing all my comments in your reply.

We are happy to learn that all concerns raised by the reviewer in the previous round were addressed adequately. We agree with the reviewer that there are several ways the work can be improved.

The various points raised by the reviewers at weaknesses are desired to be taken up in future works.